# Deep sampling of Hawaiian *Caenorhabditis elegans* reveals high genetic diversity and admixture with global populations

Tim A Crombie[1], Stefan Zdraljevic[1,2], Daniel E Cook[1,2], Robyn E Tanny[1], Shannon C Brady[1,2], Ye Wang[1], Kathryn S Evans[1,2], Steffen Hahnel[1], Daehan Lee[1], Briana C Rodriguez[1], Gaotian Zhang[1], Joost van der Zwagg[1], Karin Kiontke[3], Erik C Andersen[1]*

[1]Department of Molecular Biosciences, Northwestern University, Evanston, United States; [2]Interdisciplinary Biological Sciences Program, Northwestern University, Evanston, United States; [3]Department of Biology, New York University, New York, United States

**Abstract** Hawaiian isolates of the nematode species *Caenorhabditis elegans* have long been known to harbor genetic diversity greater than the rest of the worldwide population, but this observation was supported by only a small number of wild strains. To better characterize the niche and genetic diversity of Hawaiian *C. elegans* and other *Caenorhabditis* species, we sampled different substrates and niches across the Hawaiian islands. We identified hundreds of new *Caenorhabditis* strains from known species and a new species, *Caenorhabditis oiwi*. Hawaiian *C. elegans* are found in cooler climates at high elevations but are not associated with any specific substrate, as compared to other *Caenorhabditis* species. Surprisingly, admixture analysis revealed evidence of shared ancestry between some Hawaiian and non-Hawaiian *C. elegans* strains. We suggest that the deep diversity we observed in Hawaii might represent patterns of ancestral genetic diversity in the *C. elegans* species before human influence.

*For correspondence:
erik.andersen@northwestern.edu

**Competing interests:** The authors declare that no competing interests exist.

## Introduction

Over the last 50 years, the nematode *Caenorhabditis elegans* has been central to many important discoveries in the fields of developmental, cellular, and molecular biology. The vast majority of these insights came from the study of a single laboratory-adapted strain collected in Bristol, England known as N2 (*Brenner, 1974*; *Chalfie et al., 1994*; *Consortium TCES, 1998*; *Fire et al., 1998*; *Grishok et al., 2000*; *Hodgkin and Brenner, 1977*; *Lee et al., 1993*; *Sulston et al., 1983*). Recent sampling efforts have led to the identification of numerous wild *C. elegans* strains and enabled the study of genetic diversity and ecology of the species (*Andersen et al., 2012*; *Barrière and Félix, 2014*; *Cook et al., 2016*; *Félix and Duveau, 2012*; *Ferrari et al., 2017*; *Hahnel et al., 2018*; *Lee et al., 2019*; *Richaud et al., 2018*). The earliest studies of *C. elegans* genetic variation showed that patterns of single-nucleotide variant (SNV) diversity were shared among most wild strains, with the exception of a Hawaiian strain, CB4856, which has distinct and high levels of variation relative to other strains (*Cutter, 2006*; *Koch et al., 2000*; *Rockman and Kruglyak, 2009*). Subsequent analyses revealed that *C. elegans* has reduced levels of diversity relative to the obligate outcrossing *Caenorhabditis* species and the facultative selfer *C. briggsae* (*Dey et al., 2013*; *Thomas et al., 2015*). The most comprehensive analysis of *C. elegans* genetic diversity to date used data from thousands of genome fragments across a globally distributed collection of 97 genetically distinct strains to show

that recent selective sweeps have largely homogenized the genome (*Andersen et al., 2012*). The authors hypothesized that these selective sweeps might contain alleles that facilitate human-assisted dispersal and/or increase fitness in human-associated habitats. Consistent with the previous analyses, two Hawaiian strains, CB4856 and DL238, did not share patterns of reduced genetic diversity caused by the selective sweeps that affected the rest of the *C. elegans* population – a trend that has held true as the number of Hawaiian strains has increased (*Cook et al., 2017*; *Cook et al., 2016*; *Hahnel et al., 2018*; *Lee et al., 2019*). Taken together, these studies suggest that the Hawaiian *C. elegan*s population might be more representative of ancestral genetic diversity that existed prior to the selective pressures associated with recent human influence.

To better characterize the genetic diversity of the *C. elegans* species on the Hawaiian Islands, we performed deep sampling across five Hawaiian islands: Kauai, Oahu, Molokai, Maui, and the Big Island. Because incomplete data on locations and environmental parameters are common issues for some field studies of *C. elegans* (*Andersen et al., 2012*; *McGrath et al., 2009*; *Rockman and Kruglyak, 2009*), we developed a standardized collection procedure with the Fulcrum mobile data collection application. This streamlined procedure enabled us to rapidly record GPS coordinates and environmental niche parameters at each collection site, and accurately link these data with the nematodes we isolated. The Hawaiian Islands are an ideal location to study characteristics of the *C. elegans* niche because the Islands contain many steep, wide-ranging gradients of temperature, humidity, elevation, and landscape usage. In total, we collected samples from 2263 sites across the islands and isolated 2531 nematodes, including 309 individuals from the *Caenorhabditis* genus. Among these isolates, we identified 100 new *C. elegans* strains, 95 of which proliferated in the lab and were whole-genome sequenced. Analysis of genomic variation revealed that these strains represent 26 distinct genome-wide haplotypes not sampled previously. We refer to these genome-wide haplotypes as isotypes. We grouped these 26 Hawaiian isotypes with the 17 previously isolated Hawaiian isotypes and compared their genetic variation to 233 non-Hawaiian isotypes from around the globe. Consistent with previous observations, we found that the Hawaiian isotypes had approximately three times more diversity than the non-Hawaiian isotypes. However, we were surprised to find that, in a subset of Hawaiian isotypes, some genomic regions appear to be shared with non-Hawaiian isotypes from around the globe. These results provide the first evidence of gene flow between these populations and suggest that future sampling efforts in the Hawaiian Islands and the surrounding Pacific region will help elucidate the evolutionary processes that have shaped the genetic diversity in the *C. elegans* species.

## Results

### Hawaiian nematode diversity

In August 2017, we collected a total of 2263 samples across five Hawaiian islands and ascertained the presence of nematodes in each sample (*Figure 1*, *Supplementary file 1*, *Source data 1*). We isolated one or more nematodes from 1120 of 2263 (49%) samples, and an additional 431 of 2263 (19%) samples had circumstantial evidence of nematodes (tracks were observed but no nematodes could be isolated). Altogether, we isolated 2531 nematodes from 1120 samples and genotyped them by analysis of the Internal Transcribed Spacer (ITS2) region between the 5.8S and 28S rDNA genes (*Barrière and Félix, 2014*; *Kiontke et al., 2011*). We refer to isolates where the ITS2 region was amplified by PCR as 'PCR-positive', isolates with no amplification as 'PCR-negative', and isolates from which we could not extract high-quality genomic DNA for PCR as 'Not genotyped' (see Materials and methods). The PCR-positive category comprises *Caenorhabditis* isolates that we identified to the species level and isolates from genera other than *Caenorhabditis* that we identified to the genus level. Using this categorization strategy, we found that 427 of 2531 isolates (17%) were PCR-positive and belonged to 13 distinct taxa. Among all isolates, we identified five *Caenorhabditis* species at different frequencies across the 2263 samples: *C. briggsae* (4.2%), *C. elegans* (1.7%), *C. tropicalis* (0.57%), *C. kamaaina* (0.088%), and a new species *C. oiwi* (0.53%) (*Source data 1*). We formally describe a new species (Appendix 1), which we named *Caenorhabditis oiwi* for the Hawaiian word meaning 'native' in reference to its endemic status on the Hawaiian Islands. This species was found to be distinct based on molecular barcodes (*Kiontke et al., 2011*) and on biological species inference from crosses (*Félix et al., 2014*). *C. briggsae* was the most common *Caenorhabditis*

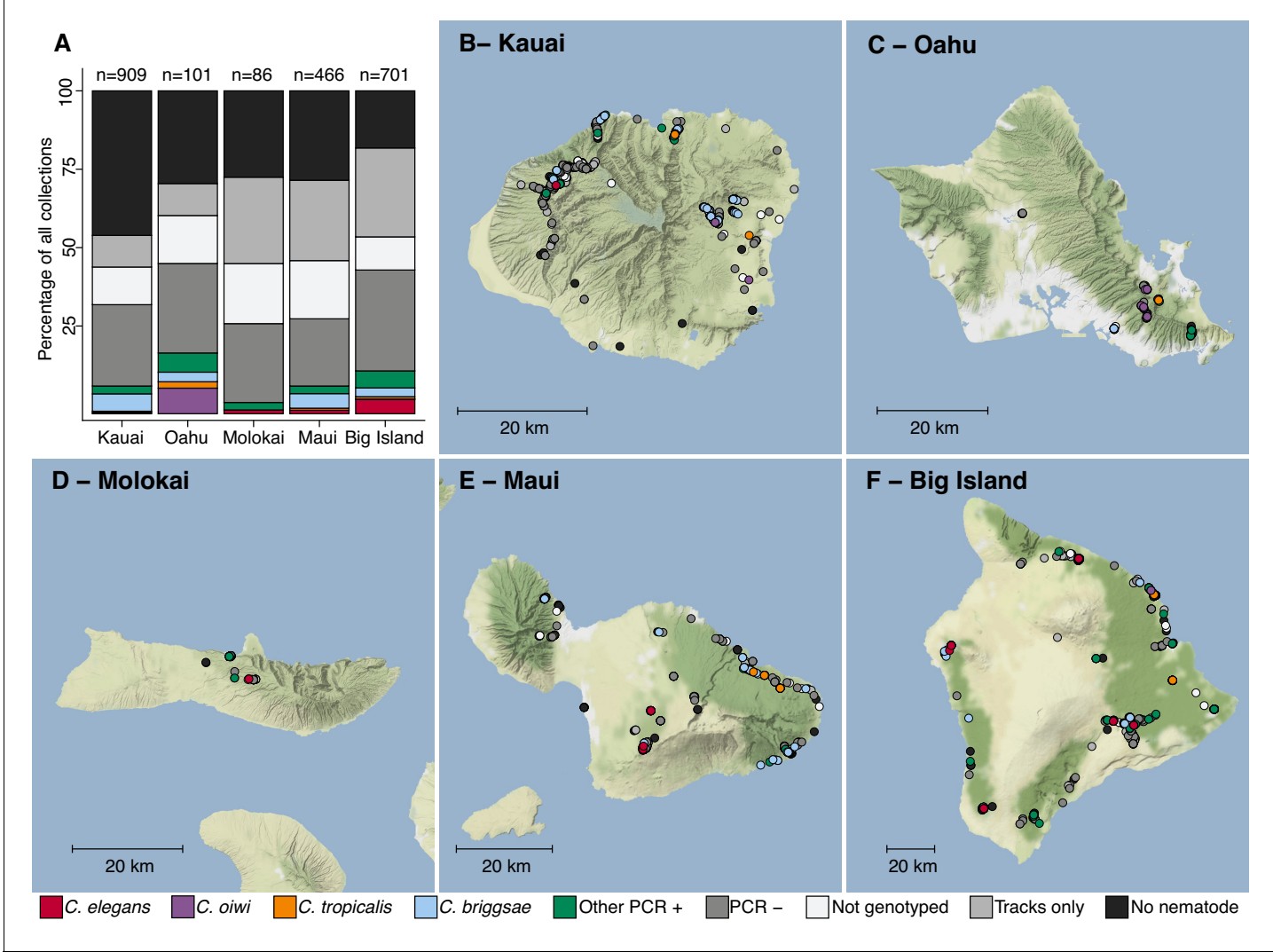

**Figure 1.** Geographic distribution of sampling sites across five Hawaiian islands. In total we sampled 2263 unique sites. (A) The percentage of each collection category is shown by island. The collection categories are colored according to the legend at the bottom of the panel, and the total number of samples for each island are shown above the bars. (B–F) The circles indicate unique sampling sites (n = 2,263) and are colored by the collection categories shown in the bottom legend. For sampling sites where multiple collection categories apply (n = 299), the site is colored by the collection category shown in the legend from left to right, respectively. For all sampling sites, the GPS coordinates and collection categories found at that site are included in (*Source data 1*). We focused our studies on *Caenorhabditis* nematode collections, excluding *C. kamaaina* because it was only found at two sampling sites.

The online version of this article includes the following source data and figure supplement(s) for figure 1:

**Source data 1.** The collection categories and location data for each of the 2263 samples collected are organized here.
**Figure supplement 1.** DIC micrographs of *C. oiwi* sp. n.
**Figure supplement 2.** Features of the male tail of *C.oiwi* sp. n.

species we isolated, which is consistent with nematode collection efforts by other groups that suggest *C. briggsae* is a ubiquitous species in many regions of the world (*Félix et al., 2013*). We found that two *Caenorhabditis* species, *C. elegans* and *C. oiwi*, were enriched on certain islands. *C. elegans* was enriched on the Big Island relative to Kauai and Maui (Fisher's Exact Test, p<0.01), and *C. oiwi* was enriched on Oahu relative to Kauai, Maui, and the Big Island (Fisher's Exact Test, p<0.00001).

## The *C. elegans* niche is distinct from other *Caenorhabditis* species on Hawaii

To characterize more about a nematode niche on the Hawaiian Islands, we classified the substrate for each distinct collection and measured various environmental parameters at the collection sites. Of the six major classes of substrate, we found nematodes most often on leaf litter (55%) (*Figure 2A*). When we accounted for collections with nematode-like tracks on the collection plate, we estimated that greater than 80% of leaf litter substrates contained nematodes (*Figure 2A*). The isolation success rate for the other classes of substrate ranged from 14% to 49% (*Figure 2A*). In comparison to overall nematode isolation rates, *Caenorhabditis* nematodes were isolated more frequently from flower substrates (40 of 202 collections) than any other substrate category (Fisher's

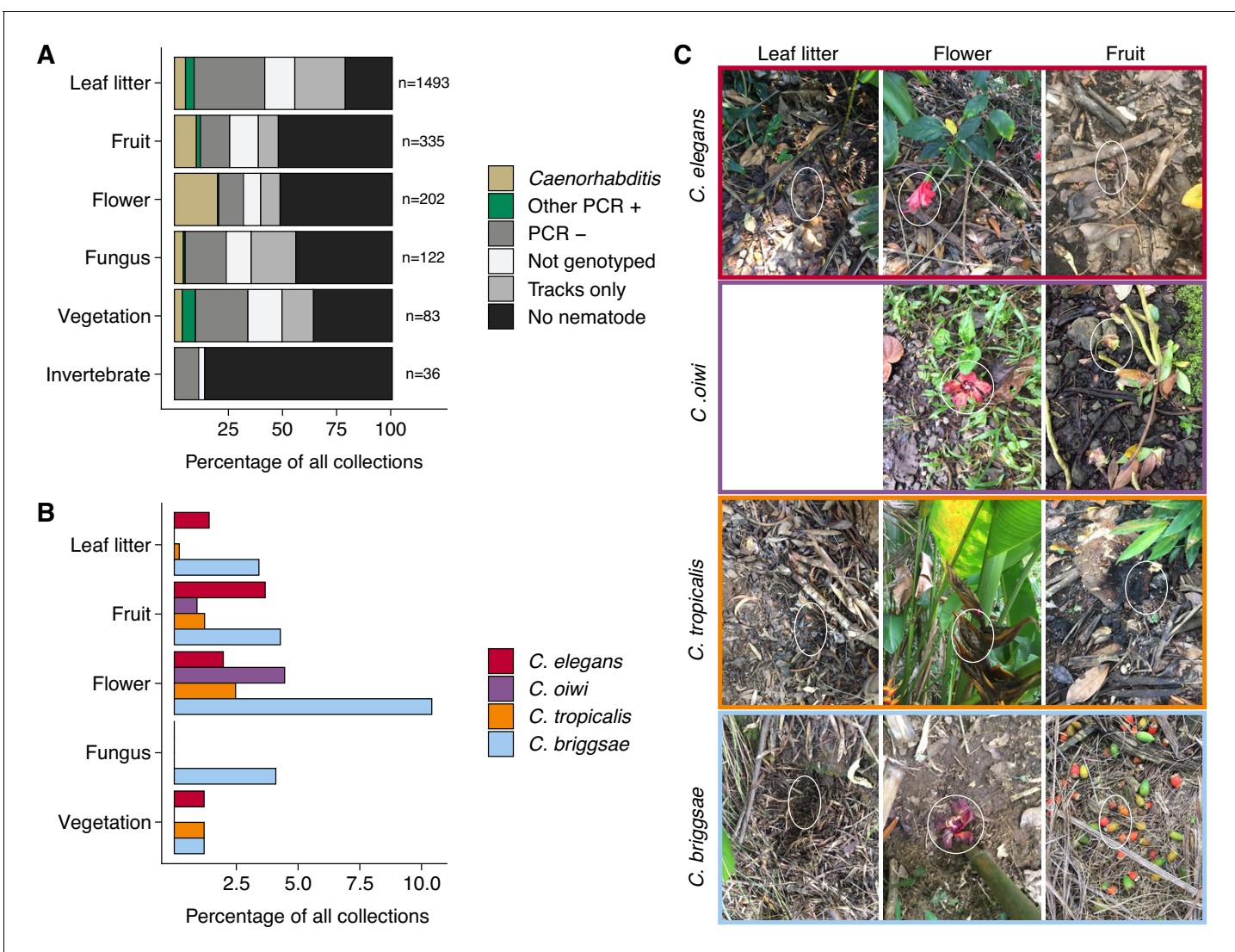

**Figure 2.** Collection categories by substrate type. (A) The percentage of each collection category is shown by substrate type. The collection categories are colored according to the legend at the right, and the total number of samples for each substrate are shown to the right of bars. (B) The percentage of collections is shown by substrate type for each *Caenorhabditis* species (excluding *C. kamaaina*, n = 2). (C) Examples of substrate photographs for *Caenorhabditis* species are shown and white ellipses indicate what was sampled. The *C. oiwi* leaf litter cell is blank because *C. oiwi* was only isolated from flowers and fruit.

The online version of this article includes the following source data for figure 2:

**Source data 1.** The fraction of samples in each collection category organized by major substrate classes.
**Source data 2.** The fraction of samples containing each *Caenorhabditis* species by substrate class are organized here.

Exact Test, p<0.026) (*Figure 2A*). We also found that *Caenorhabditis* nematodes were enriched on fruit (33 of 327 collections) relative to leaf litter substrates (76 of 1493 collections) (Fisher's Exact Test, p<0.011) but not other substrate classes (*Figure 2A*). These findings are consistent with other collection surveys that have shown leaf litter substrates harbor fewer *Caenorhabditis* nematodes than rotting flowers and fruits (*Félix et al., 2013*; *Ferrari et al., 2017*). We observed similar trends of flower-substrate enrichment relative to leaf litter for *C. briggsae* (Fisher's Exact Test, p=0.00044; flower, 21 of 202 collections and leaf litter 51 of 1493 collections) and *C. tropicalis* (Fisher's Exact Test, p=0.0056; flower, five of 202 collections and leaf litter, three of 1493 collections) but not for *C. elegans* (Fisher's Exact Test, p=1), which exhibited no substrate enrichment (*Figure 2B–C*). Interestingly, the new species, *C. oiwi*, was only isolated from flower and fruit and was enriched on flower substrates (Fisher's Exact Test, p=0.013; flower, nine of 202 collections; and fruit, three of 327 collections) (*Figure 2B–C*).

The enrichment of *C. briggsae*, *C. tropicalis*, and *C. oiwi* on flowers might indicate that this substrate class has a higher nutrient quality for these species. If this hypothesis is correct, we might expect to see a greater incidence of proliferating populations on flower substrates than other substrates. However, we saw no observable association between large population size (approximate number of nematodes on collection plate) and substrate class for *C. briggsae* (Spearman's $rho = -0.0198$, p=0.566 flower vs. leaf litter), *C. tropicalis* (Spearman's $rho = -0.258$, p=0.732 flower vs. leaf litter), nor *C. oiwi* (Spearman's $rho = 0.258$, p=0.209 flower vs. fruit), which suggests that other factors might drive the observed flower enrichment or that we are limited by the small sample size. Taken together, these data suggest that the *Caenorhabditis* species we isolated do not exhibit substrate specificity, despite flower-substrate preferences of *C. briggsae*, *C. tropicalis*, and *C. oiwi*, which is different from some other species in the genus that demonstrate substrate specificity (*e.g.*, *C. astrocarya* and *C. inopinata*) (*Ferrari et al., 2017*; *Kanzaki et al., 2018*).

In addition to recording substrate classes, we measured elevation, ambient temperature and humidity, and substrate temperature and moisture to determine if these niche parameters were important for individual *Caenorhabditis* species (*Figure 3*; see Materials and methods). Consistent with previous *C. elegans* collections in tropical regions (*Andersen et al., 2012*; *Dolgin et al., 2008*), all *C. elegans* isolates were collected from elevations greater than 500 meters and were generally found at higher elevations than other *Caenorhabditis* species (*Figure 3E*; mean = 867 m; elevation: Dunn test, p<0.00001). We also found that *C. elegans*-positive collections tended to be at cooler ambient and substrate temperatures than other *Caenorhabditis* species (ambient temperature: Dunn test, p<0.005; substrate temperature: Dunn test, p<0.00001), although these two environmental parameters were correlated with elevation (*Figure 3F*). Notably, the average substrate temperatures for *C. elegans* (19.4˚C), *C. tropicalis* (26.0˚C), and *C. briggsae* (23.7˚C) positive collections are close to the optimal growth temperatures for these species in the laboratory setting (*Figure 3B*) (*Poullet et al., 2015*). Our collections also indicate that *C. oiwi* tends to be found on drier substrates than *C. elegans* (*Figure 3D*; Dunn test, p=0.0021), but we observed no differences among species for ambient humidity (*Figure 3C*). Given the similar substrate and environmental parameter preferences of *C. tropicalis*, *C. briggsae*, and *C. oiwi*, we next asked if these species colocalized at either the local (<30 m$^2$) or substrate (<10 cm$^2$) scales. To sample at the local scale, we collected samples from 20 gridsects (see Materials and methods; *Figure 3—figure supplement 1*) and observed no colocalization of these three species, although only 16% of the total collections were a part of a gridsect. At the substrate scale, we found *C. tropicalis* and *C. briggsae* cohabiting on two of 108 substrates with either species present, and *C. oiwi* and *C. briggsae* cohabiting on one of 107 substrates with either species present (*Figure 3—figure supplement 2*). Among 95 substrates with *C. briggsae*, we observed nine instances of *C. briggsae* cohabiting with other PCR-positive species. We did not collect any samples that harbored *C. elegans* and any other *Caenorhabditis* species. We emphasize that these co-occurrence frequencies represent a lower bound estimate of the real co-occurrence frequencies, because in many cases we only isolated a small fraction of the nematodes present on a particular sample. Taken together, these results highlight the ubiquitous nature of *C. briggsae* on the Hawaiian Islands and further suggest that the niche of *C. elegans* might be distinct from *C. tropicalis*, *C. briggsae*, and *C. oiwi* on the Hawaiian Islands.

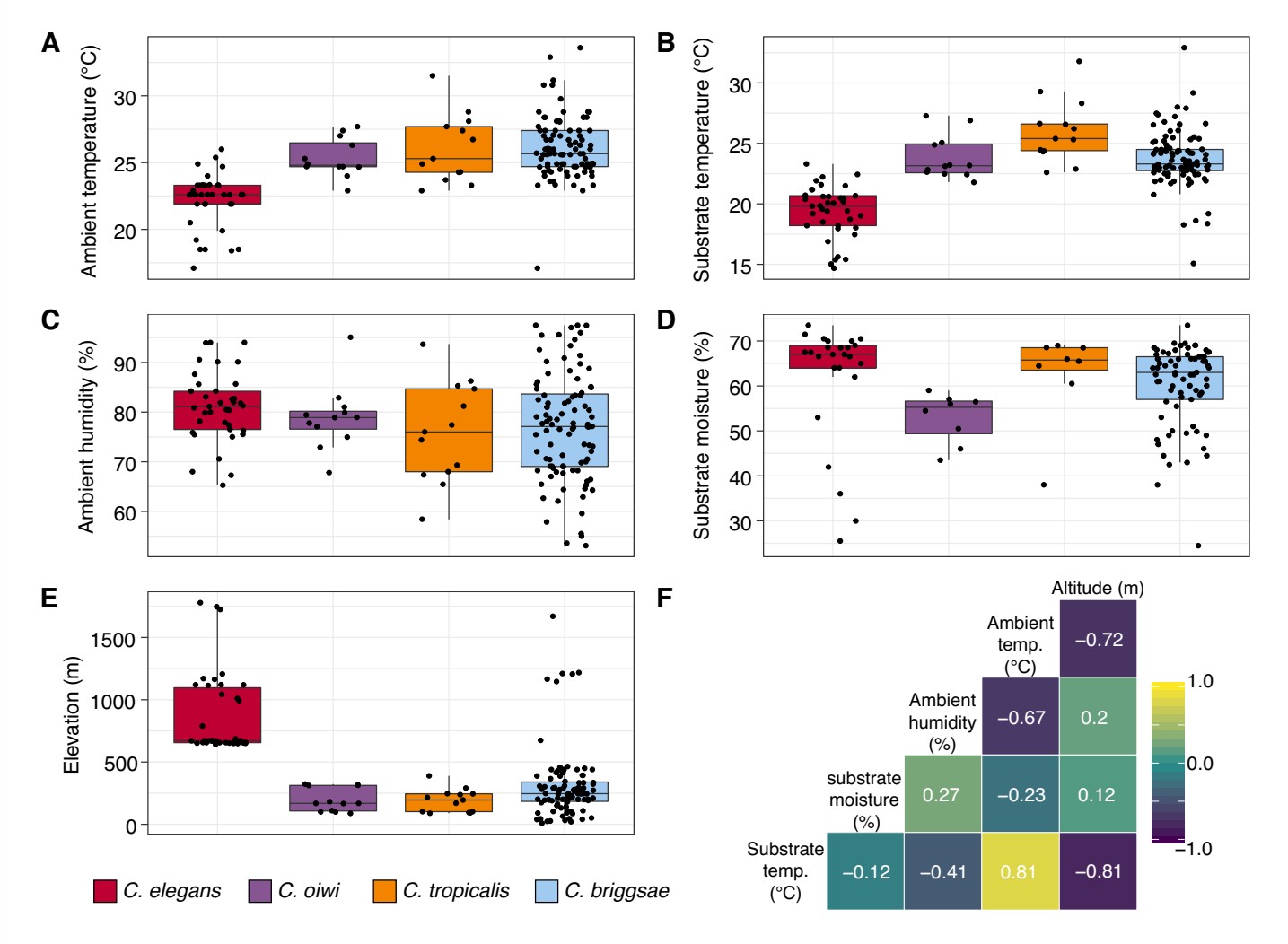

**Figure 3.** Environmental parameter values for sites where *Caenorhabditis* species were isolated. (A–E) Tukey box plots are plotted by species (colors) for different environmental parameters. Each dot corresponds to a unique sampling site where that species was identified. In cases where two *Caenorhabditis* species were identified from the same sample (n = 3), the same parameter values are plotted for both species. All p-values were calculated using Kruskal-Wallis test and Dunn test for multiple comparisons with *p* values adjusted using the Bonferroni method; comparisons not mentioned were not significant (α = 0.05). (A) Ambient temperature (˚C) was typically cooler at the sites were *C. elegans* were isolated compared to sites for all other *Caenorhabditis* species (Dunn test, p<0.005). (B) Substrate temperature (˚C) was also generally cooler for *C. elegans* than all other *Caenorhabditis* species (Dunn test, p<0.00001). (C) Ambient humidity (%) did not differ significantly among the *Caenorhabditis*-positive sites. (D) Substrate moisture (%) was generally greater for *C. elegans* than *C. oiwi* (Dunn test, p=0.002). (E) Elevation (meters) was typically greater at sites where *C. elegans* were isolated compared to sites for all other *Caenorhabditis* species (Dunn test, p<0.00001). (F) A correlation matrix for the environmental parameters was made using sample data from the *Caenorhabditis* species shown. The parameter labels for the matrix are printed on the diagonal, and the Pearson correlation coefficients are printed in the cells. The color scale also indicates the strength and sign of the correlations shown in the matrix. The online version of this article includes the following source data and figure supplement(s) for figure 3:

**Source data 1.** Environmental parameter data for all *Caenorhabditis*-positive collections are organized here.

**Source data 2.** Pearson correlation coefficients for environmental parameter pairs.

**Figure supplement 1.** Local scale sampling with gridsects.

**Figure supplement 1—source data 1.** Collection data for all gridsect samples.

**Figure supplement 2.** Network of cohabiting species isolated from samples.

**Figure supplement 2—source data 1.** Collection data for all samples where two or more PCR-positive nematodes were isolated from the same sample.

**Figure supplement 3.** Local diversity and colocalization of isotypes.

**Figure supplement 3—source data 1.** Collection data for gridsects from which *C. elegans* were isolated.

**Figure supplement 3—source data 2.** Collection data for all samples from which multiple *C. elegans* isotypes were isolated.

# Hawaiian *C. elegans* are divergent from most strains sampled across the globe

We previously showed that two *C. elegans* isolates from Hawaii are highly divergent relative to wild isolates from other regions of the world and represent a large portion of the genetic diversity found within the species (*Andersen et al., 2012*). Since this analysis, an additional 15 isolates have been collected from the islands and show similarly high levels of genetic diversity (*Cook et al., 2016*; *Hahnel et al., 2018*). To better characterize the genetic diversity in Hawaii, we acquired whole-genome sequence data from 95 *C. elegans* isolates that we collected in this study. By analyzing the variant composition of these 95 isolates, we identified 26 distinct genome-wide haplotypes that we refer to as isotypes (see Materials and methods). Within these 26 isotypes, we identified approximately 1.54 million single nucleotide variants (SNVs) that passed our filtering strategy (see Materials and methods; hard-filter VCF; *Supplementary file 2*), which is 27.6% greater than the total number of SNVs identified in all of the 233 non-Hawaiian isotypes included in this study. We found that distinct isotypes are frequently isolated within close proximity to one another in Hawaii. We identified up to seven unique isotypes colocalized within a single gridsect (less than 30 m$^2$) (*Figure 3—figure supplement 3A*). We also found that colocalization occurred at the substrate level; among the 38 substrates from which we isolated *C. elegans*, 12 contained two or more isotypes (*Figure 3—figure supplement 3B*).

The variant data from all 43 Hawaiian isotypes (26 new with 17 previously described Hawaiian isotypes) allowed us to perform detailed analyses of Hawaiian genetic diversity. Consistent with what is known about *C. elegans* genetic diversity (*Andersen et al., 2012*), we observed a high degree of genome-wide relatedness among the majority of non-Hawaiian isotypes (*Figure 4—figure supplement 1*). By contrast, isotypes sampled from Hawaii are divergent from isolates sampled outside of Hawaii with the exception of five non-Hawaiian isotypes. Among these exceptions, ECA36 and QX1211 were collected from urban gardens in New Zealand and San Francisco, CA, respectively, and grouped with some of the most divergent isotypes from Hawaii. The other exceptions were the isotypes JU2879, MY23, and MY16, which were isolated across the Pacific region, (*i.e.*, Mexico City, Mexico; Concepción, Chile; Nantou County, Taiwan) as well as near the city of Münster, Germany (*Petersen et al., 2015*). This result suggests that high genetic diversity is not specific to the Hawaiian Islands but might also be common throughout the Pacific region. Within the sampled Hawaiian isotypes, genome-wide relatedness revealed a high degree of divergence (*Figure 4—figure supplement 1*). This trend is further supported by elevated levels of genome-wide average nucleotide diversity (π) in the Hawaiian sample relative to the non-Hawaiian sample, which we found to be three-fold higher (Hawaii π = 0.00109; non-Hawaiian π = 0.000368; *Figure 4A*). The genomic distribution of diversity followed a similar pattern across chromosomes for both samples, wherein chromosome centers and tips exhibited lower diversity on average than chromosome arms (*Figure 4A*). This pattern is likely explained by lower recombination rates, higher gene densities, and elevated levels of background selection on chromosome centers (*Consortium TCES, 1998*; *Cutter and Payseur, 2003*; *Rockman et al., 2010*). Interestingly, we observed discrete peaks of diversity in specific genomic regions (*e.g.*, chr IV center), which suggests that balancing selection might maintain diversity at these loci in both samples (*Figure 4A*). This hypothesis is supported by corresponding spikes in Tajima's *D* (*Figure 4B*) (*Tajima, 1989*). Alternatively, higher values of Tajima's *D* might indicate a population contraction, but the discrete nature of these peaks makes this possibility less likely. A third possible explanation is that uncharacterized structural variation (*e.g.*, duplication and divergence) exists in these regions. Nevertheless, the variant sites within these discrete peaks in π and Tajima's *D* are unlikely the result of sequencing errors because they are identified across multiple isotypes (see Materials and methods). Our previous analysis showed that 70–90% of isotypes contain reduced levels of diversity across several megabases (Mb) on chromosomes I, IV, V, and X (*Andersen et al., 2012*). This reduced diversity was hypothesized to be caused by selective sweeps that occurred within the last few hundred years, potentially through drastic alterations of global environments by humans. The two Hawaiian isotypes, CB4856 and DL238, did not share this pattern of reduced diversity, suggesting that they avoided the selective pressure. Consistent with this previous analysis, we did not observe signatures of selection in the sampled Hawaiian isotypes on chromosomes I, IV, V, and X, as measured by Tajima's *D* (*Figure 4B*), which suggests that the Hawaiian and non-Hawaiian samples have distinct evolutionary histories. This distinction is

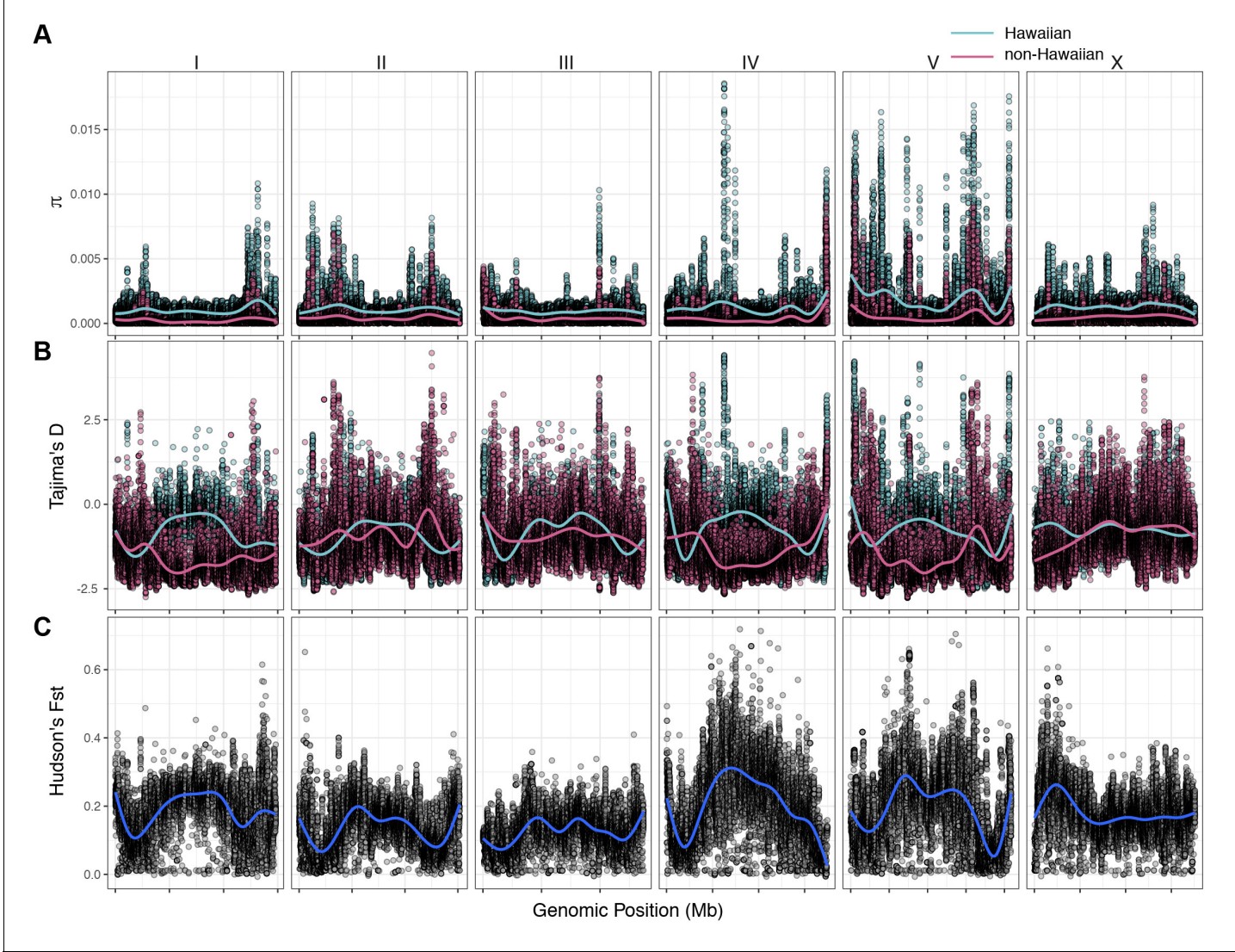

**Figure 4.** Chromosomal patterns of *C. elegans* diversity and differentiation. All comparisons are between the 43 Hawaiian isotypes and the 233 non-Hawaiian isotypes from the rest of the world. All statistics were calculated along a sliding window of size 10 kb with a step size of 1 kb. Each dot corresponds to the calculated value for a particular window. (A) Genome-wide $\pi$ calculated for Hawaiian isotypes (light blue) and non-Hawaiian isotypes (pink) are shown. (B) Genome-wide Tajima's *D* statistics for Hawaiian isotypes (light blue) and non-Hawaiian isotypes (pink) are shown. (C) Genome-wide Hudson's $F_{ST}$ comparing the Hawaiian and non-Hawaiian isotypes are shown.

The online version of this article includes the following source data and figure supplement(s) for figure 4:

**Source data 1.** Chromosomal patterns of *C. elegans* diversity and differentiation.
**Source data 2.** Chromosomal patterns of Tajima's *D* for the Hawaiian and non-Hawaiian samples.
**Figure supplement 1.** *Caenorhabditis elegans* unrooted tree for 276 isotypes.
**Figure supplement 1—source data 1.** Best-scoring maximum likelihood tree output from RAxML-ng.

also captured in genome-wide Hudson's $F_{ST}$, where the differentiation between the two samples is highest in regions of the genome impacted by the selective sweeps (*Figure 4C*) (*Bhatia et al., 2013*; *Hudson et al., 1992*). Taken together, these data suggest that the isotypes sampled from Hawaii have largely been insulated from the selective pressures thought to be associated with human activity in many regions of the world.

## *C. elegans* population structure

To assess population structure among all 276 isotypes sampled, we performed admixture analysis (see Materials and methods). This analysis identified at least 11 genetic groups for this set of *C. elegans* isotypes, each of which we refer to as a population for the purposes of further analyses (*Figure 5*). Support for at least 11 distinct populations (K) is shown by the minimization of cross-validation (CV) error between Ks 11–15 (*Figure 5—figure supplement 1*). Furthermore, the population assignments for K = 11 closely aligned to the relatedness clusters we observed in a neighbor-joining network of all Hawaiian isotypes and the species-wide tree (*Figure 5*, *Figure 4—figure supplement 1*). For Ks 11–15, the majority of Hawaiian isotypes consistently exhibit no admixture with non-Hawaiian populations. However, a minority of Hawaiian isotypes are consistently either admixed with non-Hawaiian populations (*e.g.* K = 11, 14, and 15) or assigned to populations that contain non-Hawaiian isotypes (*e.g.* K = 12 and 13) (*Figure 5—figure supplement 1*). These data support that a subset of Hawaiian isotypes consistently exhibit a greater degree of genetic relatedness with non-Hawaiian isotypes across different numbers of population subdivisions (Ks). Together, we found at least four distinct populations sampled from the Hawaiian Islands and at least seven additional non-Hawaiian populations sampled from around the globe (*Figure 5—figure supplement 2*).

The majority of isotypes assigned to the seven non-Hawaiian populations exhibit a high degree of admixture with one another, indicating that these populations are not well differentiated. By contrast, isotypes assigned to three of the four Hawaiian populations showed little admixture with other populations (*Figure 5—figure supplement 2A*). We refer to the four Hawaiian populations as Volcano, Hawaiian Divergent, Hawaiian Invaded, and Hawaiian Low for the following reasons. All eight isotypes in the Volcano population were isolated on the Big Island of Hawaii from the town of Volcano at high elevation in wet rainforests primarily composed of ferns, 'Ōhi'a lehua, and koa trees. The Hawaiian Divergent population is named for the two highly divergent isotypes, XZ1516 and ECA701, which were isolated from Kauai, the oldest Hawaiian island sampled. However, we emphasize that the population assignment of these two highly divergent isotypes might not be correct given that they each contain many strain-specific variants that were filtered from the admixture analysis. The Hawaiian Invaded population is named because many of the isotypes assigned to this population exhibited admixture with non-Hawaiian populations, which is suggestive of an invasion of non-Hawaiian alleles into Hawaii (*Figure 6A*, *Figure 6—figure supplement 1*, *Figure 6—figure supplement 2*). The Hawaiian Low population is named because isotypes assigned to this population tended to be isolated at lower elevations than those assigned to the other Hawaiian populations (See Materials and methods, *Figure 6B*). Together, our data suggest that the population structure of the Hawaiian isotypes is associated with either Island, elevation, or both.

Within the Hawaiian Invaded population, one of the 19 isotypes was isolated from outside of Hawaii (MY23), and 11 of 18 Hawaiian isotypes showed admixture with various non-Hawaiian populations, particularly the non-Hawaiian population C (*Figure 6*, *Figure 5—figure supplement 2*). By contrast, just one isotype (QG556, from the C population) was admixed with the Hawaiian Invaded population and it was isolated from California (*Figure 6—figure supplement 1*). This result suggested that these populations either share ancestry or recent gene flow occurred between them. To distinguish between these possibilities, we explicitly tested for the presence of gene flow among all populations using TreeMix (*Pickrell and Pritchard, 2012*), which estimates the historical relationships among populations accounting for both population splits and migration events. We found evidence of gene flow between the Hawaiian Invaded population and the non-Hawaiian population C (*Figure 7*; *Figure 7—figure supplement 1*). The topological position of the fourth highest-weight migration event identified by TreeMix (*i.e.*, C→Hawaiian Invaded) suggested that the evidence of gene flow is not caused by incomplete assortment of ancestral alleles (*i.e.*, the migration arrows connect the 'C' and Hawaiian Invaded lineages at the branch tips) (*Figure 7C*; *Figure 7—figure supplement 1*). TreeMix also detected evidence of gene flow from a population related to the Hawaiian Low and Hawaiian Invaded populations to the non-Hawaiian G population (*Figure 7C*; *Figure 7—figure supplement 1*). The G population comprises 15 isotypes typically isolated from Portugal and its two archipelagic autonomous regions, the Azores and Madeira. Interestingly, all of the G isotypes collected from the Azores and Madeira (JU258, NIC199, NIC251) are greater than 13% admixed with the Hawaii Low population (*Figure 5—figure supplement 2A*). Although we do not know exactly where the source population for this migration was located, the relatively high genetic

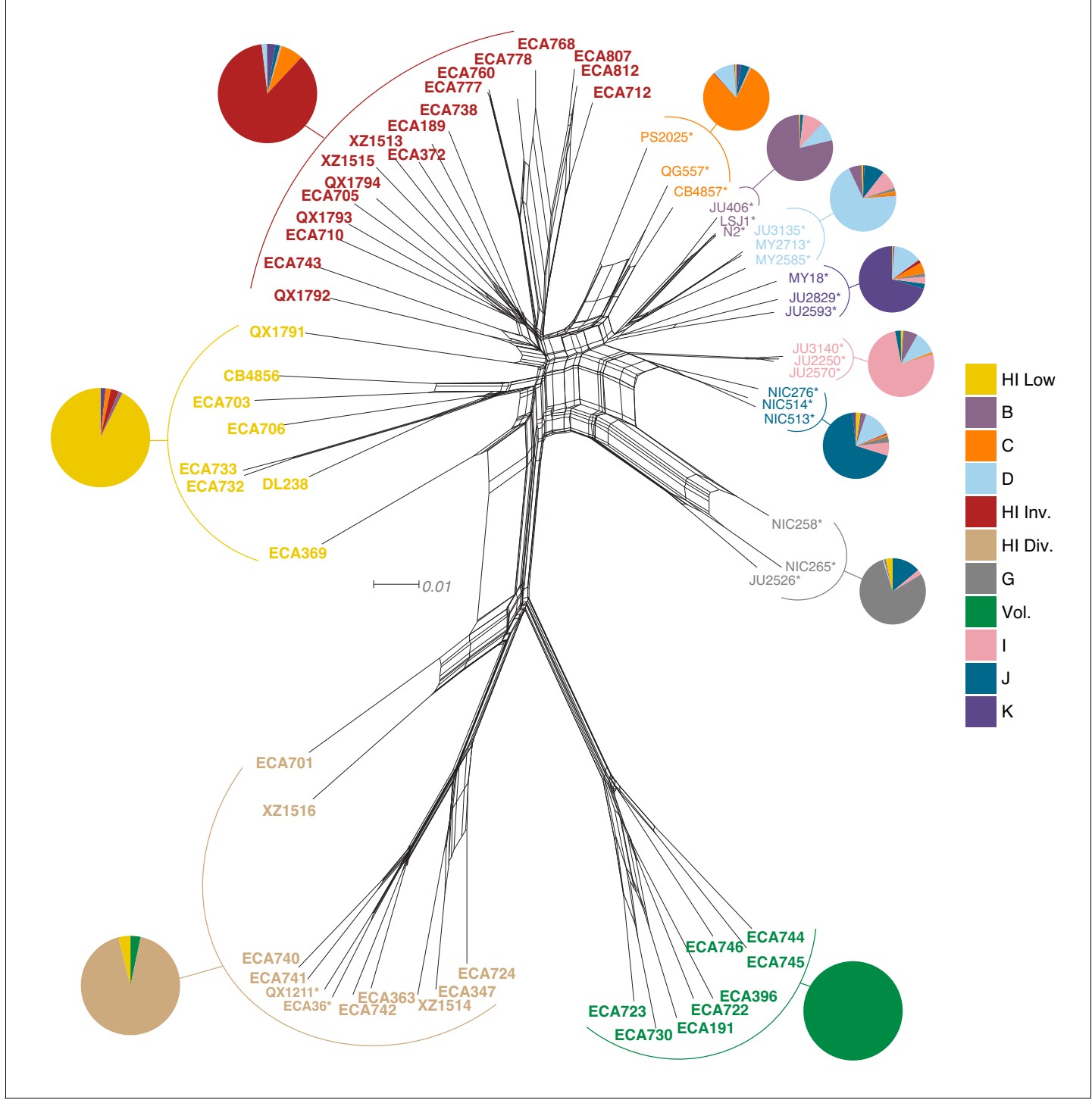

**Figure 5.** Relatedness of the Hawaiian *C. elegans* isotypes. Neighbor-joining net showing the genetic relatedness of the Hawaiian *C. elegans* isotypes relative to a representative set of non-admixed, non-Hawaiian isotypes from each population defined by ADMIXTURE (K = 11). Colors of isotype names indicate the maximum fraction of population assignment from ADMIXTURE (K = 11), including the seven non-Hawaiian populations (**B–K**) and the four Hawaiian populations (Hawaiian Invaded, Hawaiian Low, Hawaiian Divergent, and Volcano). Isotypes labeled with an asterisk are representative of non-admixed, non-Hawaiian isotypes from each population defined by ADMIXTURE (K = 11). Pie charts represent population proportions for all isotypes within the full admixture population.

The online version of this article includes the following source data and figure supplement(s) for figure 5:

**Source data 1.** VCF used to generate the nexus file with vcf2phylip.py script for the neighbor-joining network shown in *Figure 5*.
**Source data 2.** The nexus file for creating neighbor-joining network shown in *Figure 5*.

*Figure 5 continued on next page*

*Figure 5 continued*

**Source data 3.** Admixture population proportions for all isotypes within each admixture population (K = 11).

**Figure supplement 1.** Summary of ADMIXTURE analysis.

**Figure supplement 1—source data 1.** Cross-validation error from ten Independent ADMIXTURE runs at Ks 2–20.

**Figure supplement 1—source data 2.** The inferred population proportions estimated by ADMIXTURE for all isotypes at Ks 10–15.

**Figure supplement 2.** Global population structure.

**Figure supplement 2—source data 1.** The sampling locations and admixture population assignment for all strains including isotype reference strains.

**Figure supplement 2—source data 2.** The inferred population proportions estimated by ADMIXTURE for all isotypes at K = 11.

**Figure supplement 3.** Seasonal temperature variation by population.

**Figure supplement 3—source data 1.** Seasonal temperature variation for all isotypes over a 12-month period.

**Figure supplement 4.** Summary of ADMIXTURE analysis on Hawaiian *C. elegans* isotypes.

**Figure supplement 4—source data 1.** The inferred population proportions estimated by ADMIXTURE for all Hawaiian isotypes at Ks 3–5.

**Figure supplement 4—source data 2.** Cross-validation error from ten Independent ADMIXTURE runs with only the Hawiian isotypes across Ks 2–10.

**Figure supplement 4—source data 3.** The nexus file for creating the neighbor-joining network.

---

diversity sampled within the Pacific region suggests that it was likely located there and possibly within Hawaii. Importantly, TreeMix cannot definitively distinguish the direction of migration between populations.

To further assess evidence of gene flow between the Hawaiian and non-Hawaiian populations, we analyzed the haplotype structure across the genomes of all 276 *C. elegans* isotypes (*Browning and Browning, 2016*). Within the Hawaiian Divergent, Volcano, and Hawaiian Low populations, we observed haplotypes that were largely absent from the non-Hawaiian isotypes. By contrast, the Hawaiian Invaded population shared portions of haplotypes that were commonly found in non-Hawaiian populations. For example, the isotypes in the Hawaiian Invaded population exhibiting admixture with the non-Hawaiian C population share haplotype arrangements on the left and center of Chr III (red and orange Chr III, *Figure 7A*). We also found evidence of the globally swept haplotype in all of the isotypes from the Hawaiian Invaded population, particularly on chromosomes I, V, and X, but less so on chromosome IV (*Figure 7A–B*, *Figure 7—figure supplement 2*). By contrast, greater than 50% of chromosome IV contained the swept haplotype in all of the isotypes from the C population (*Figure 7—figure supplement 2*). The Hawaiian Low population generally contained smaller portions of the globally swept haplotypes than the Hawaii Invaded population with the exception of the QX1791 isotype, which is likely because of its admixture with non-Hawaiian populations 'C' and 'K' (*Figure 7A–B*, *Figure 5—figure supplement 2A*). Taken together, our data showed that the Hawaiian isotypes from the Volcano and Hawaiian Divergent populations have avoided the selective sweeps that are found in isotypes across most regions of the globe, and individuals within the Hawaiian Invaded subpopulation have likely been outcrossed with these swept haplotypes.

## Discussion

We sought to deeply sample the natural genetic variation within the *C. elegans* species to better understand the evolutionary history and driving forces of genome evolution in this powerful model system. Because the Hawaiian Islands have been shown to harbor highly divergent strains relative to most regions of the world, we choose to sample extensively on these islands. We developed a streamlined collection procedure that facilitated our collection of over 2000 samples across five Hawaiian islands. From these collections, we isolated over 2500 nematodes and used molecular data to partition 427 of these isolates into 13 distinct taxa, mostly from the Rhabditidae family. In total, we identified and cryogenically preserved 95 new *C. elegans* isolates that represent 26 genetically distinct isotypes. These isotypes represent the largest single *C. elegans* collection effort on any island system and contain 27% more SNVs than all 233 non-Hawaiian isotypes combined. Our findings confirm previous results where high diversity is found on the Hawaiian Islands (*Andersen et al., 2012*), and document the first evidence of gene flow between Hawaiian and non-Hawaiian populations. Additionally, the wealth of genetic variation within these new Hawaiian isotypes can be used as a source of novel variants that might impact phenotypes of interest. It is important to note that future genome-wide association mappings using these highly diverged isotypes, which contain population-specific SNVs, might detect spurious associations that cannot be mitigated easily by

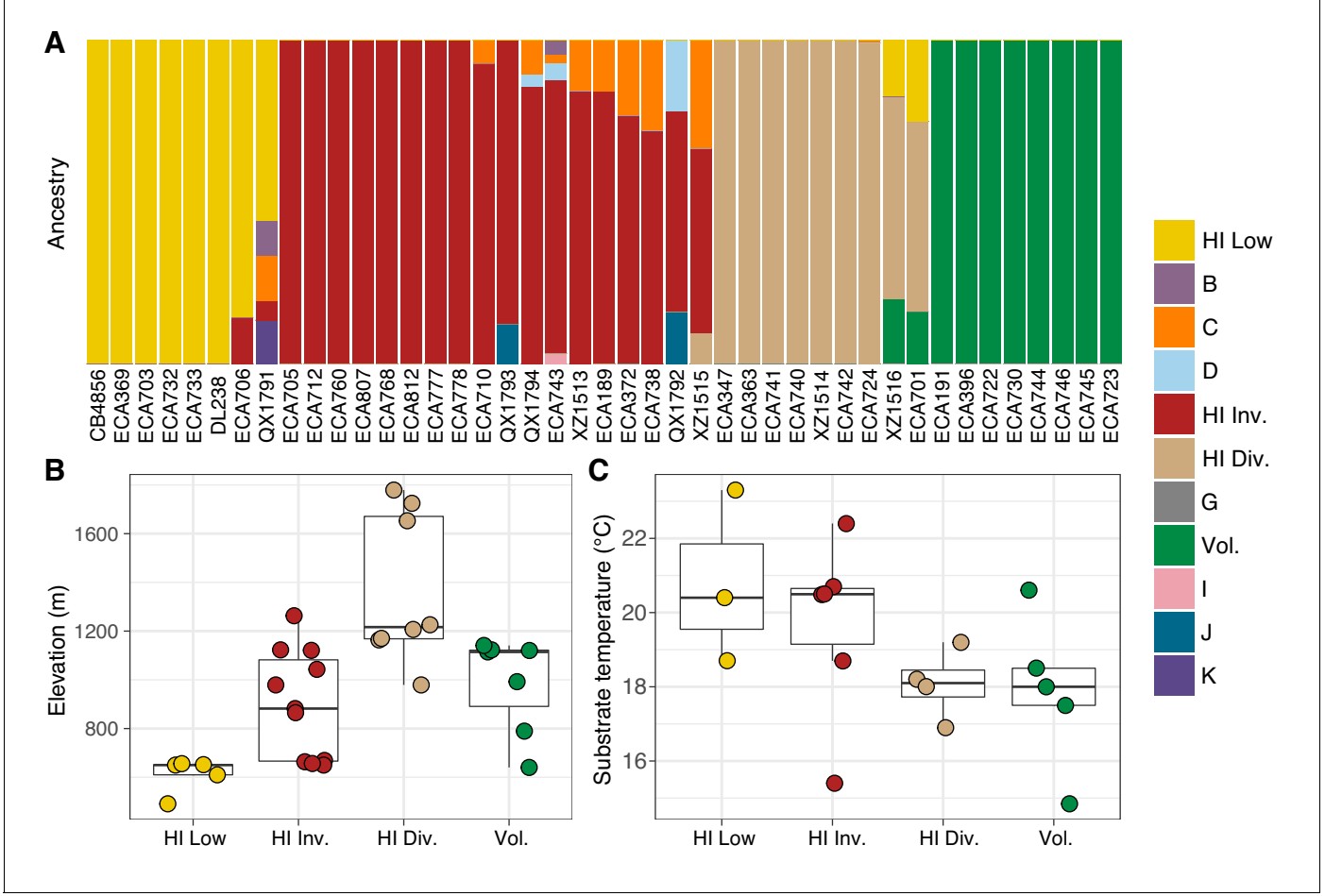

**Figure 6.** Environmental parameters of Hawaiian *C. elegans* populations. (**A**) The inferred ancestral population fractions for each Hawaiian isotype as estimated by ADMIXTURE (K = 11; 276 *C. elegans* isotypes) are shown. The bar colors represent the fraction of population assignments from ADMIXTURE for the isotypes named on the x-axis. (**B–C**) Tukey box plots are shown by population assignments (colors) for different environmental parameters. We used the average values of environmental parameters from geographically clustered collections to avoid biasing our results by local oversampling (See Materials and methods - Environmental parameter analysis). All p-values were calculated using Kruskal-Wallis test and Dunn test for multiple comparisons with *p* values adjusted using the Bonferroni method; comparisons not mentioned were not significant (α = 0.05). (**B**) The collection site elevations for Hawaiian isotypes colored by population assignments are shown. The Hawaiian Low and the Hawaiian Invaded populations were typically found at lower elevations than the Hawaiian Divergent population (Dunn test, p-values=0.000168, and 0.037 respectively). (**C**) The substrate temperatures for Hawaiian isotypes colored by population assignments are shown.

The online version of this article includes the following source data and figure supplement(s) for figure 6:

**Source data 1.** The inferred population fractions for each Hawaiian isotype as estimated by ADMIXTURE (K = 11; 276 *C. elegans* isotypes).

**Source data 2.** Environmental parameters for distinct collections of admixture populations in Hawaii.

**Source data 3.** *post hoc* Dunn multiple comparison tests for differences in environmental parameters among the Hawaiian populations identified by ADMIXTURE are organized here.

**Figure supplement 1.** Admixture between the Hawaii Invaded and the global C populations.

**Figure supplement 1—source data 1.** Isolation location data for distinct isolations of all isotypes and their inferred population proportions estimated by ADMIXTURE.

**Figure supplement 2.** Admixture between the 'Hawaiian Invaded and 'global C' populations.

**Figure supplement 2—source data 1.** The inferred population proportions estimated by ADMIXTURE (K = 11) for isotypes assigned to the Hawaii Invaded, Volcano, and non-Hawaiian C populations.

corrections of population stratification (*Zhou and Stephens, 2012*). Like other diverse population samples, further sampling and a larger number of individuals will help facilitate these types of mappings in the future.

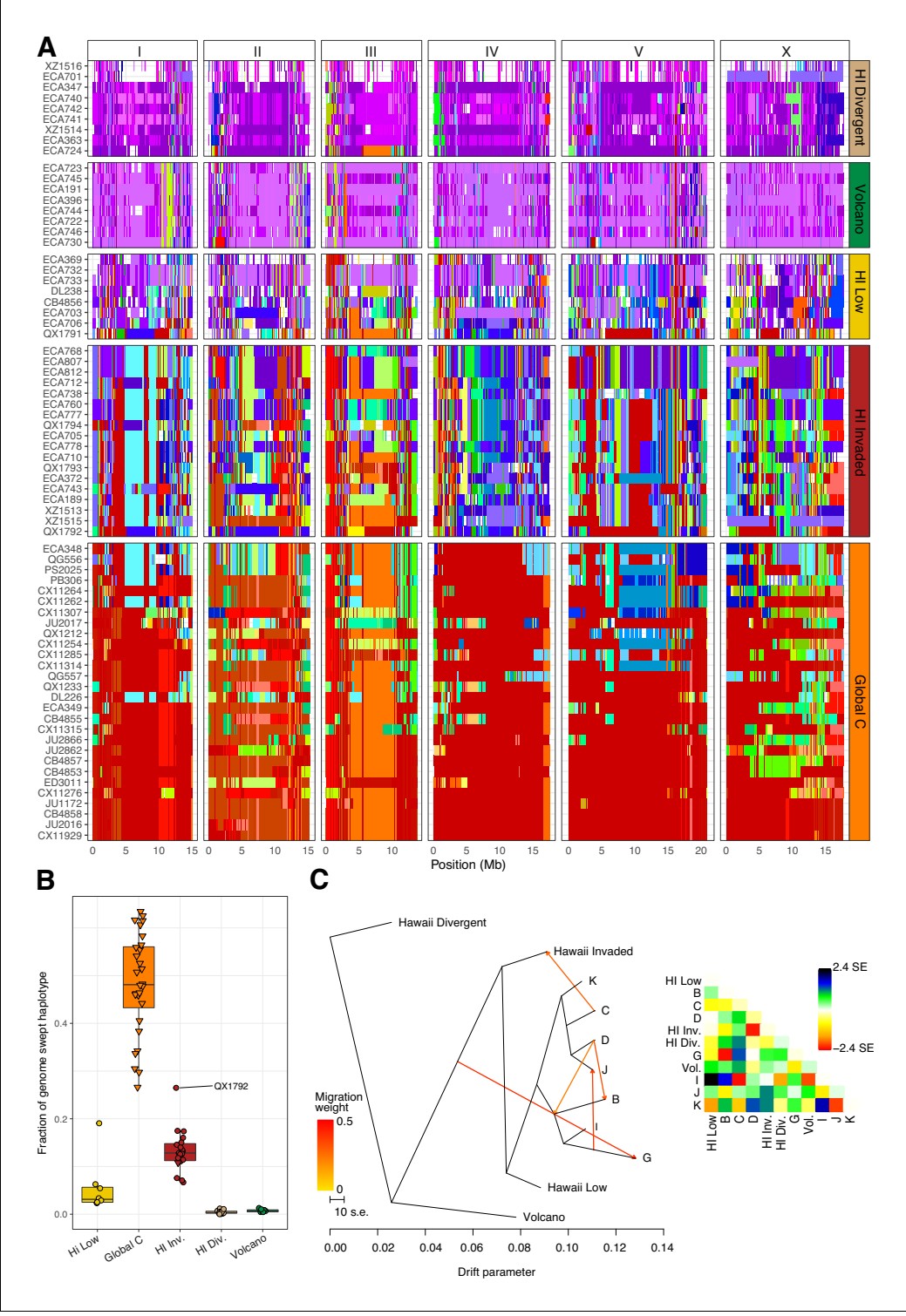

**Figure 7.** Evidence of migration between the Hawaiian and world populations. (A) The haplotypes or inferred blocks of identity by descent (IBD) across the genome are shown. The genomic position is plotted on the x-axis for each isotype plotted on the y-axis. The block colors correspond to a uniquely defined IBD group. The dark red blocks correspond to the most common global haplotype (*i.e.*, the swept haplotypes on chr I, IV, V, and left of X). Genomic regions with no color represent regions for which no IBD groups could be determined. The four Hawaiian populations are shown in the top four facets, excluding non-Hawaiian isotypes. The bottom facet shows the non-Hawaiian C population. (B) The total fraction of the genome with the swept haplotype is shown by

*Figure 7 continued on next page*

*Figure 7 continued*

population. The data points correspond to isotypes and are colored by their assigned populations. The Hawaiian isotypes are plotted as circles and non-Hawaiian isotypes are plotted as triangles. Hawaiian isotypes with greater than 25% of their genome swept are labelled. (C) The inferred relationship among the populations allowing for five migration events (ADMIXTURE, K = 11). The heat map to the right represents the residual fit to the migration model.

The online version of this article includes the following source data and figure supplement(s) for figure 7:

**Source data 1.** Haplotypes or IBD blocks across the genomes of all 276 isotypes are organized here.
**Source data 2.** Fraction of the genome with the swept haplotypes for each isotype in the four Hawaiian populations and the non-Hawaiian C population.
**Source data 3.** Ancestry fractions (.Q file) and allele frequencies of the inferred ancestral populations (.P file) from ADMIXTURE analysis.
**Figure supplement 1.** Evidence of migration between Hawaiian and non-Hawaiian populations.
**Figure supplement 1—source data 1.** Ancestry fractions (.Q file) and allele frequencies of the inferred ancestral populations (.P file) from ADMIXTURE analysis (K = 11).
**Figure supplement 2.** Most common global haplotype sharing by chromosome and population.
**Figure supplement 2—source data 1.** Fraction of each chromosome with the most common global haplotype (swept haplotypes for chromosomes I, IV, V, and X) for each isotype in the four Hawaiian populations and the non-Hawaiian C population.

## The origins of *C. elegans*

The origin of many species can be traced to the location where genetic diversity is highest (*Li and Stephan, 2006*; *Nielsen et al., 2017*; *Peter et al., 2018*). The high genetic diversity we sampled from Hawaii suggests that the Hawaiian Islands could be the origin of *C. elegans*. The extant Hawaii Islands range in age from the still-forming Big Island to the 5.1 million year old Kauai, but the now submerged Emperor Seamounts represent approximately 70 million years of stable land masses over the Pacific Hotspot (*Neall and Trewick, 2008*). These older landmasses predate the split between *C. elegans* and its closest know relative *C. inopinata,* which is estimated to be 10.5 million years (*Kanzaki et al., 2018*). However, divergent isolates are also found throughout the Pacific region and sampling efforts historically are biased towards Europe. Notably, *C. inopinata*, was isolated in South Japan, suggesting that these species could have originated from a common ancestor located outside of Hawaii but within the Pacific region (*Kanzaki et al., 2018*). If *C. elegans* did originate from the Pacific region or the Hawaiian Islands, it is possible that the high diversity sampled from these locations could have been maintained after the transition from a genetically divergent outcrossing species to a selfing species (*Thompson et al., 2015*). Outcrossing *Caenorhabditis* species generally harbor much higher levels of genetic diversity as compared to selfing *Caenorhabditis* species (*Graustein et al., 2002*; *Jovelin et al., 2003*; *Li et al., 2014*; *Cutter et al., 2019*). Nevertheless, the high diversity among the Hawaiian isotypes, all of which are selfing hermaphroditic strains, implies that the origin of selfing might have occurred earlier in the *C. elegans* evolutionary history than previously hypothesized (*Cutter et al., 2008*).

The higher genetic diversity sampled from the Hawaii Islands could have at least two possible explanations other than residual ancestral diversity present since the transition to selfing. First, the higher genetic diversity in the Hawaiian Islands might be driven by population demography on the Islands. It is possible that Hawaii harbors larger, more temporally stable effective population sizes than other regions of the world that have been sampled. Under a neutral model, populations with a larger effective population size are expected to have a greater number of neutral polymorphisms (*Kimura, 1991*). These larger, more stable effective population sizes are plausible in Hawaii given the abundant supply of available habitats (*e.g.* rotting fruits and vegetation) and stable temperatures throughout the year. The Hawaiian climate is particularly less variable than many temperate regions where *C. elegans* populations are known to exhibit seasonal population expansions and contractions (*Frézal and Félix, 2015*; *Richaud et al., 2018*). Furthermore, population bottlenecks associated with long-range dispersal to new habitats might also contribute to reduced diversity in many parts of the globe. Second, high genetic diversity might not have been restricted to Hawaii but present throughout the globe until recently. If true, then recent selective sweeps associated with human activity might have purged diversity from most regions except the Hawaiian Islands. This explanation is less

plausible because genetically diverse isotypes are rarely found outside of the Pacific region, despite massive sampling efforts (especially in Europe). Ultimately, the pattern of genetic variation in Hawaiian populations is likely influenced by a combination of demographic history (*e.g.*, reproductive mode, changes in population size, short- and long-range migration events, and admixture) as well as evolutionary processes such as natural selection, recombination, and mutation. To further untangle the evolutionary history and origin of this species, additional samples from natural areas around the globe and in particular the Pacific region will be required.

## Out of Hawaii or invasion of Hawaii?

Our sampling efforts within the Hawaiian Islands allowed us to explore how gene flow into and out of the Hawaiian Islands contributes to the patterns of genetic diversity we observe within the current global sample of *C. elegans*. In particular, we asked whether haplotype sharing among Hawaiian and non-Hawaiian isotypes could be explained by emigration of nematodes from Hawaii (out of Hawaii) or immigration of nematodes to Hawaii (invasion of Hawaii).

Our data support the immigration of alleles into the Hawaiian Invaded population from the less-diverse, non-Hawaiian C population. Moreover, most isotypes from the Hawaiian Invaded and non-Hawaiian populations share portions of swept haplotypes. Although, isotypes within the Hawaiian Invaded population contain smaller portions of the swept haplotypes than those sampled outside of Hawaii. This pattern of haplotype sharing could be explained by an 'invasion of Hawaii' scenario, wherein swept haplotypes have invaded Hawaii. The genetic diversity of the Hawaiian populations might be threatened if the invading alleles confer strong fitness advantages as is expected for swept haplotypes (*Andersen et al., 2012*). However, if an invasion of Hawaii is currently underway, we have little evidence to support the selection of the swept haplotypes in Hawaii. First, the Hawaiian Invaded population only contains small fractions of the swept haplotypes on chromosomes I, V, X, and even smaller fractions on chromosome IV. Second, it would take a large number of generations to create these small swept haplotypes because of the low outcrossing rates and high incidence of outbreeding depression in *C. elegans* (*Dolgin et al., 2007*). Therefore, we speculate that the swept haplotypes introgressed into the Hawaiian Invaded population do not confer the same fitness advantages in the high elevation, cool temperature Hawaiian niche as they do in other parts of the world.

We also discovered evidence suggesting that alleles have emigrated from Hawaii to other regions of the globe. TreeMix analyses suggests that gene flow occurred from a population related to the Hawaiian Invaded and Hawaiian Low populations into the non-Hawaiian G population. Although we cannot be certain if this source population was located within Hawaii, its relation to the two Hawaiian populations suggests that it was. This 'Out of Hawaii' scenario might have occurred through long-range dispersal of genotypes similar to those found in the Hawaiian Invaded and Hawaiian Low populations to other regions of the globe. Recent migration out of Hawaii could have been aided by the transition of the Hawaiian economy towards large-scale production and export of sugarcane and tropical fruits, which began in the late nineteenth century (*Bartholomew et al., 2012*). If correct, then this situation is similar to what is thought to have occurred within *Drosophila melanogaster* where the fruit trade might have facilitated recent migrations from native regions to oceanic islands (*David and Capy, 1988*; *Hales et al., 2015*). Taken together, our results suggest that gene flow into and out of the Hawaiian Islands contributes greatly to the patterns of genetic diversity we observe in *C. elegans* today.

## The ancestral niche of *C. elegans* might be similar to the Hawaiian niche

We used publicly available weather data from the National Oceanic and Atmospheric Administration and the National Climatic Data Center to measure the variation in seasonal temperatures for locations close to the sites where isotypes were collected (*Evans et al., 2017*). We found that the Hawaiian sampling locations experienced less seasonal variability in temperature than any of the non-Hawaiian locations (*Figure 5—figure supplement 3*). These findings raise the possibility that the ancestral niche of *C. elegans* might be similar to the thermally stable Hawaiian habitats where genetic diversity is highest. However, factors other than seasonal temperature variation might also characterize the ancestral niche of *C. elegans*. The Hawaiian Divergent population was enriched at higher elevations, which have been less impacted by human activities in Hawaii since the time of Polynesian colonization (*Alison Kay, 1994*). By contrast, the Hawaiian Invaded and the Hawaiian Low

populations are found at lower elevations, each of which show evidence of the swept haplotypes. Although it remains unclear what factors restrict the flow of swept haplotypes to the Hawaiian Divergent and Volcano populations, selective pressures connected to reduced human influence might contribute to the lack of genetic exchange. This possibility would be consistent with the hypothesis that the global sweeps, present in the Hawaiian Invaded and Hawaiian Low populations, originated through positive selection acting on loci that confer fitness advantages in human-associated habitats (*Andersen et al., 2012*). Alternatively, the presence of swept haplotypes in the Hawaiian Invaded and Hawaiian Low populations may simply reflect that they are found at lower elevations closer to the coasts where they might be more likely to mate with immigrating nematodes. Nevertheless, we suspect that the ancestral niche of *C. elegans* is most likely to be similar to the thermally stable, high elevation Hawaiian habitats where human impacts are less prevalent.

## Unravelling the evolutionary history of *C. elegans*

More accurate models of *C. elegans* niche preferences will facilitate our ability to unravel the evolutionary history of this species by directing researchers to areas most likely to harbor *C. elegans* populations. In order to build more accurate niche models, future sampling efforts should include unbiased sampling across environmental gradients in multiple locations over time because data on niche parameters where *C. elegans* is not found is as important as data where *C. elegans* is found (*Félix and Duveau, 2012*; *Ferrari et al., 2017*; *Petersen et al., 2014*; *Richaud et al., 2018*). Additionally, we must identify and quantify important biotic niche factors, including associated bacteria, fungi, and invertebrates. These types of data will help facilitate the identification of genes and molecular processes that are under selection in different subpopulations across the species range. *C. elegans* offers a tractable and powerful animal model system to connect environmental parameters to functional genomic variation. These data will deepen our understanding of the evolutionary history of *C. elegans* by revealing how selection and demographic forces have shaped the genome of this important model system.

# Materials and methods

Key resources table

| Reagent type (species) or resource | Designation | Source or reference | Identifiers | Additional information |
|---|---|---|---|---|
| Sequence-based reagent | oECA305; ITS2 F primer | This paper | Andersen Lab: oECA305 F primer | GCTGCGTTATTTACCACGAATTGCARAC |
| Sequence-based reagent | oECA202; ITS2 R primer | 10.1186/1471-2148-11-339 | Andersen Lab: oECA202 R primer | GCGGTATTTGCTACTACCAYYAMGATCTGC |
| Sequence-based reagent | RHAB1350F; Rhabditid F primer | 10.1093/molbev/msh264 | Andersen Lab: oECA1271 | TACAATGGAAGGCAGCAGGC |
| Sequence-based reagent | RHAB1868R; Rhabditid R primer | 10.1093/molbev/msh264 | Andersen Lab: oECA1272 | CCTCTGACTTTCGTTCTTGATTAA |
| Commercial assay or kit | Blood and Tissue DNA isolation kit | QIAGEN | cat# 69506 | |
| Commercial assay or kit | Qubit dsDNA Broad Range Assay Kit | Invitrogen | cat# Q32850 | |
| Software, algorithm | Fulcrum | Spatial Networks | Spatial Networks: Fulcrum | https://www.fulcrumapp.com/ |
| Software, algorithm | UGENE | 10.1093/bioinformatics/bts091 | Unipro: UGENE v.1.27.0 | http://ugene.net/ |
| Software, algorithm | Nextflow | 10.1038/nbt.3820 | Nexflow | https://github.com/nextflow-io/nextflow |

## Strains

Nematodes were reared at 20°C using *Escherichia coli* OP50 bacteria grown on modified nematode growth medium (NGMA), containing 1% agar and 0.7% agarose to prevent animals from burrowing

(*Andersen et al., 2014*). In total, 169 *C. briggsae*, 100 *C. elegans*, 21 *C. tropicalis*, 15 *C. oiwi*, and four *C. kamaaina* wild isolates were collected. Of these strains, 95 *C. elegans,* 19 *C. tropicalis*, and 12 *C. oiwi* wild isolates were cryopreserved and are available upon request along with the other *C. elegans* strains included in our analysis (*Source data 2*). The type specimen for *C. oiwi* (ECA1100) is also deposited at the *Caenorhabditis* Genetics Center (Appendix 1).

## Sampling strategy

We sampled nematodes at 2263 sites across five Hawaiian Islands during August 2017. Before travelling to Hawaii, general sampling locations were selected based on accessibility via hiking trails and by proximity to where *C. elegans* had been collected previously (*Andersen et al., 2012*; *Cook et al., 2016*; *Hahnel et al., 2018*; *Hodgkin and Doniach, 1997*). Sampling hikes with large elevation changes were prioritized to ensure that we sampled across a broad range of environmental parameters. On these hikes, we opportunistically sampled substrates known to harbor *C. elegans*, including fruits, flowers, stems, leaf litter, fungus, compost, wood, live arthropods, and molluscs (*Ferrari et al., 2017*; *Frézal and Félix, 2015*; *Schulenburg and Félix, 2017*). We chose to group these substrates into six major categories: leaf litter, fruit, flower, fungus, invertebrate, and vegetation. The vegetation category contains all substrates that do not fall into any of the other five categories. We chose to distinguish the leaf litter, fruit, and flower substrates from the vegetation category because we suspect these substrates provide distinct microhabitats and each substrate was sampled extensively. In 20 locations, we performed extensive local sampling in an approximately 30 square meter area that we refer to as a 'gridsect'. The gridsect comprised a center sampling point with additional sampling sites at one, two, and three meters away from the center in six directions with each direction 60° apart from each other (*Figure 3—figure supplement 1*).

## Field sampling and environmental data collection

To characterize the *Caenorhabditis* abiotic niche, we collected and organized data for several environmental parameters at each sampling site using a customizable geographic data-collection application called Fulcrum. We named our customized Fulcrum application 'Nematode field sampling' and used the following workflow to enter the environmental data into the application while in the field. First, we used a mobile device camera to scan a unique collection barcode from a pre-labelled plastic collection bag. This barcode is referred to as a collection label or 'C-label' in the application and is used to associate a particular sample with its environmental and nematode isolation data. Next, we entered the substrate type, landscape, and sky view data into the application using drop down menus and photographed the sample in place using a mobile device camera. The GPS coordinates for the sample are automatically recorded in the photo metadata. We then measured the surface temperature of the sample using an infrared thermometer Lasergrip 1080 (Etekcity, Anaheim, CA), its moisture content using a handheld pin-type wood moisture meter MD912 (Dr. Meter, Los Angeles, CA), and the ambient temperature and humidity near the sample using a combined thermometer and hygrometer device GM1362 (GoerTek, Weifang, China). These measurements were entered into the appropriate fields in the application (*Supplementary file 3*). Finally, we transferred the sample into a collection bag and stored it in a cool location before we attempted to isolate nematodes. Seventy samples in our raw data had missing GPS coordinates or GPS coordinates that were distant from actual sampling locations after visual inspection using satellite imagery. The positions for these samples were corrected using the average position of the two samples collected before and after the errant data point or by manually assigning estimated positions.

## Nematode isolation

Following each collection, the substrate sample was transferred from the barcoded collection bag to an identically barcoded 10 cm NGMA plate seeded with OP50 bacteria. For 1989 of the 2263 samples collected, we isolated nematodes that crawled off the substrates onto the collection plates approximately 47 hr after the samples were collected from the field (mean = 46.9 hr, std. dev. = 19.5 hr). The remaining 274 samples were shipped overnight from Hawaii to Northwestern University in collection bags, and the nematodes were isolated approximately 172 hr after sample collection (mean = 172.5 hr, std. dev. = 17.9 hr). For each collection plate, up to seven gravid nematodes were isolated by transferring them individually to pre-labeled 3.5 cm NGMA isolation plates

seeded with OP50 bacteria. We refer to these isolation plates as 'S-plates' in the Fulcrum application we called 'Nematode isolation' (*Supplementary file 4*). At the time of isolation, we recorded the approximate number of nematodes on the collection plate. We merged the collection, isolation, and environmental data together into a single data file with the 'process_fulcrum_data.R' script that can be found in the scripts folder of the GitHub repository https://github.com/AndersenLab/HawaiiMS (*Crombie, 2019*; copy archived at https://github.com/elifesciences-publications/ HawaiiMS) (*Source data 1*).

## Nematode identification

The isolated nematodes were stored at 20℃ for approximately 14 days (mean = 14.3 d, std. Dev. = 4.9 d) but were not passaged during this time to avoid multiple generations of proliferation. For initial genotyping, whenever possible five to ten nematodes were lysed in 8 µl of lysis solution (100 mM KCl, 20 mM Tris pH 8.2, 5 mM MgCl$_2$, 0.9% IGEPAL, 0.9% Tween 20, 0.02% gelatin with proteinase K added to a final concentration of 0.4 mg/ml) then frozen at −80℃ for up to 12 hr. If isolated nematodes were not found on the isolation plates prior to genotyping, they were categorized as 'Not genotyped'. If the isolation plate only contained dead or sterile nematodes, we attempted to lyse five to ten carcasses or sterile individuals. The lysed material was thawed on ice, and 1 µl was loaded directly into 40 µl reactions with primers spanning a portion of the ITS2 region (Internal Transcribed Spacer) between the 5.8S and 28S rDNA genes with forward primer oECA305 (GCTGCG TTATTTACCACGAATTGCARAC) and reverse primer oECA202 (GCGGTATTTGCTACTACCAYYA MGATCTGC) (*Kiontke et al., 2011*). The ITS2 PCR used the following conditions: 180 s denaturation step at 95℃; then 34 cycles of 95℃ for 15 s, 60℃ for 15 s, and 72℃ for two minutes; followed by a five-minute elongation step at 72℃. We also loaded 1 µl of the lysed material into 40 µl reactions with second set of primers RHAB1350F (TACAATGGAAGGCAGCAGGC) and RHAB1868R (CCTC TGACTTTCGTTCTTGATTAA), which amplify about 500 bp of 18S rDNA of Rhabditid nematodes (*Haber et al., 2005*). The Rhabditid PCR used the following conditions: 120 s denaturation step at 95℃; then 35 cycles of 95℃ for 20 s, 55℃ for 60 s, and 72℃ for 30 s; followed by a five-minute elongation step at 72℃. The presence of PCR products was visualized on a 2% agarose gel in 1X TAE buffer. Because we could not be certain of the genomic DNA template quality, isolates that did not produce the expected 500 bp Rhabditid PCR product were labelled as 'Not genotyped'. Those isolates that did not yield an ITS2 PCR product were labelled as 'PCR-negative', and those isolates that did yield the expected 2 kb ITS2 PCR product were labelled as 'PCR-positive'. To identify the genus of the isolates labeled as 'PCR-positive', we then used Sanger sequencing of the ITS2 PCR products with forward primer oECA305. We classified *Caenorhabditis* species by comparing the ITS2 sequences to the National Center for Biotechnology Information (NCBI) database using the BLAST algorithm. Isolates with sequences that aligned best to genera other than *Caenorhabditis* were only classified to the genus level.

For every isolate where the BLAST results either aligned to *C. elegans*, had an unexpectedly high number of mismatches in the center of the read, or did not match any known sequences because of poor sequence quality, we performed another independent lysis and PCR using high-quality Taq polymerase (cat# RR001C, TaKaRa) to confirm our original results. For this confirmation, we used the forward primer oECA305 and the reverse primer oECA306 (CACTTTCAAGCAACCCGAC) to sequence the confirmation ITS2 amplicon in both directions. The sequence chromatograms were then quality trimmed by eye with Unipro UGENE software (version 1.27.0) and compared to known nematode species in the NCBI sequence database using the BLAST algorithm. We used the consensus alignment of the forward and reverse reads to confirm our original results. For *C. elegans*, five of the 100 strains perished before we could confirm their identity. We also confirmed that several strains that best aligned to *C. kamaaina* shared a large number of mismatches in the center of the ITS2 amplicon, suggesting they belonged to a new species. For these strains, we performed reciprocal mating tests with *C. kamaaina* to infer the new species by the biological species concept (*Félix et al., 2014*). None of these crosses produced viable progeny, suggesting that these isolates represent a new *Caenorhabditis* species (Appendix 1).

## Illumina library construction and whole-genome sequencing

To extract DNA, we transferred nematodes from two 10 cm NGMA plates spotted with OP50 *E. coli* into a 15 ml conical tube by washing with 10 mL of M9. We then used gravity to settle animals on the bottom of the conical tube, removed the supernatant, and added 10 mL of fresh M9. We repeated this wash method three times over the course of one hour to serially dilute the *E. coli* in the M9 and allow the animals time to purge ingested *E. coli*. Genomic DNA was isolated from 100 to 300 µl nematode pellets using the Blood and Tissue DNA isolation kit cat# 69506 (QIAGEN, Valencia, CA) following established protocols (*Cook et al., 2016*). The DNA concentration was determined for each sample with the Qubit dsDNA Broad Range Assay Kit cat# Q32850 (Invitrogen, Carlsbad, CA). The DNA samples were then submitted to the Duke Center for Genomic and Computational Biology per their requirements. The Illumina library construction and sequencing were performed at Duke University using KAPA Hyper Prep kits (Kapa Biosystems, Wilmington, MA) and the Illumina NovaSeq 6000 platform (paired-end 150 bp reads). The raw sequencing reads for strains used in this project are available from the NCBI Sequence Read Archive (Project PRJNA549503).

## Variant calling

To ensure reproducible data analysis, all genomic analyses were performed using pipelines generated in the Nextflow workflow management system framework (*Di Tommaso et al., 2017*). Each Nextflow pipeline used in this study is briefly described below (*Supplementary file 5*). All pipelines follow the '*pipeline name-nf*' naming convention and full descriptions can be found on the Andersen lab dry-guide website: (http://andersenlab.org/dry-guide/pipeline-overview/).

Raw sequencing reads were trimmed using *trimmomatic-nf*, which uses trimmomatic (v0.36) (*Bolger et al., 2014*) to remove low-quality bases and adapter sequences. Following trimming, we used the *concordance-nf* pipeline to characterize *C. elegans* strains isolated in this study and previously described strains (*Cook et al., 2017*; *Cook et al., 2016*; *Hahnel et al., 2018*). The *concordance-nf* pipeline calls SNVs using the BCFtools (v.1.9) (*Li, 2011*) variant calling software. The variants are filtered by: Depth (FORMAT/DP)≥3; Mapping Quality (INFO/MQ)>40; Variant quality (QUAL) >30; (Allelic Depth (FORMAT/AD)/Num of high quality bases (FORMAT/DP)) ratio >0.5. We determined the pairwise similarity of all strains by calculating the fraction of shared SNVs. Finally, we classified two or more strains as the same isotype if they shared >99.9% SNVs. If a strain did not meet this criterion, we considered it as a unique isotype. Newly assigned isotypes were added to CeNDR (*Cook et al., 2017*).

After isotypes are assigned, we used *alignment-nf* with BWA (v0.7.17-r1188) (*Li, 2013*; *Li and Durbin, 2009*) to align trimmed sequence data for distinct isotypes to the N2 reference genome (WS245) (*Lee et al., 2018*). Next, we called SNVs using *wi-nf*, which uses the BCFtools (v.1.9) (*Li, 2011*). The *wi-nf* pipeline generates two population-wide VCFs that we refer to as the soft-filtered and hard-filtered VCFs (*Supplementary file 2*). After variant calling, a soft-filtered VCF was generated for each sample by appending the following soft-filters to variant sites: Depth (FORMAT/DP)>10; Mapping Quality (INFO/MQ)>40; Variant quality (QUAL) >10; (Allelic Depth (FORMAT/AD)/Number of high quality bases (FORMAT/DP)) ratio >0.5. These soft-filters were appended to the FT field of the VCF using *VCF-kit* (*Cook and Andersen, 2017*). Next, sample VCFs were merged using the merge utility of BCFtools (v.1.9). Once the population VCF was generated, variant sites with greater than 90% missing genotypes (high_missing) or greater than 10% heterozygosity (high_heterozygosity) were flagged. We refer to this VCF as the 'soft-filtered VCF'. To construct the 'hard-filtered VCF', we removed all variants that did not pass the filters described above. Both the soft- and hard-filtered isotype-level VCFs are available to download on the CeNDR website (version 20180527) (*Cook et al., 2017*).

We further pruned the hard-filtered VCF to contain sites with no missing genotype calls using BCFtools (v.1.9) (*Li, 2011*). The predicted variant effects were appended to the VCF using SnpEff (v 4.3) (*Cingolani et al., 2012*). We further annotated this VCF with exons, G-quartets, transcription factor binding sites, histone binding sites, miRNA binding sites, splice sites, ancestral alleles (XZ1516 set as ancestor), the genetic map position, and repetitive elements using vcfanno (v 0.2.8) (*Pedersen et al., 2016*). All annotations were obtained from WS266. We removed regions that were annotated as repetitive. We named this VCF the 'PopGen VCF' (Supplementary data 3 on GitHub

https://github.com/AndersenLab/HawaiiMS/blob/master/data/elife_files/Supplemental_Data_3.vcf.gz; *Supplementary file 2*).

## Phylogenetic analyses

We characterized the relatedness of the *C. elegans* isotypes using RAxML-ng with the GTR DNA substitution model and maximum likelihood estimation to find the parameter values that maximize the phylogenetic likelihood function, and thus provide the best explanation for the observed data (*Kozlov et al., 2019*). We pruned the 'PopGen VCF' by removing sites in high linkage disequilibrium (LD) using PLINK (v1.9) (*Chang et al., 2015*; *Purcell et al., 2007*) with the *–indep-pairwise 50 1 0.95* command. We used the vcf2phylip.py script (*Ortiz, 2019*) to convert this pruned VCF to the PHYLIP format (*Felsenstein, 1993*) required to run RAxML-ng. To construct the unrooted tree that included 276 strains (*Figure 4—figure supplement 1*), we used the GTR evolutionary model available in RAxML-ng (*Lanave et al., 1984*; *Tavaré, 1986*). This tree was visualized using the ggtree (v1.10.5) R package (*Yu et al., 2017*). To construct the neighbor-net phylogenies (*Figure 5*, *Figure 5—figure supplement 4*) we further pruned the 'PopGen VCF' by removing sites in linkage disequilibrium using PLINK (v1.9) with the `-indep-pairwise 50 1 0.8` command. We also removed variants only present in one isotype. To visualize the neighbor-net phylogenies we used SplitsTree4 (*Huson and Bryant, 2006*).

## Population genetic statistics

Genome-wide $\pi$, Hudson's $F_{ST}$, and Tajima's D were calculated using the PopGenome package in R (*Pfeifer et al., 2014*). All statistics were calculated along sliding windows with a 10 kb window size and a 1 kb step size with the 'PopGen VCF' (Supplementary data 3 on GitHub https://github.com/AndersenLab/HawaiiMS/blob/master/data/elife_files/Supplemental_Data_3.vcf.gz; *Supplementary file 2*). Importantly, our statistics are calculated among isotypes not among isolates. The standard method for calculating these statistics is among isolates. We choose to calculate these values among isotypes to avoid oversampling of highly related individuals that are often isolated from the same substrate.

## Admixture analysis

We performed admixture analysis using ADMIXTURE (v1.3.0) (*Alexander et al., 2009*). Prior to running ADMIXTURE, we LD-pruned the 'PopGen VCF' (Supplementary data 3 on GitHub https://github.com/AndersenLab/HawaiiMS/blob/master/data/elife_files/Supplemental_Data_3.vcf.gz) using PLINK (v1.9) (*Chang et al., 2015*; *Purcell et al., 2007*) with the command `-indep-pairwise 50 10 0.8`. We also removed variants only present in one isotype. We ran ADMIXTURE ten independent times for K sizes ranging from 2 to 20 for all 276 isotypes. Visualization of admixture results was performed using the pophelper (v2.2.5) R package (*Francis, 2017*). We chose K = 11 for future analyses because the cross-validation (CV) error approached minimization at this K (*Figure 5—figure supplement 1*). Furthermore, K = 11 subset the Hawaiian isotypes into four distinct populations, which exactly matched the subsets obtained from running ADMIXTURE on just the 43 Hawaiian isotypes at K = 4 (K = 4 minimized CV for ADMIXTURE with Hawaiian isotypes only, (*Figure 5—figure supplement 4*). We performed TreeMix analysis on K = 11 for zero to five migration events (*Pickrell and Pritchard, 2012*).

## Haplotype analysis

We determined identity-by-descent (IBD) of strains using IBDSeq (*Browning and Browning, 2013*) run on the 'PopGen VCF' (Supplementary Data 3) with the following parameters: `minalleles = 0.01, ibdtrim = 0, r2max = 0.8`. IBD segments were then used to infer haplotype structure among isotypes as described previously (*Andersen et al., 2012*). After haplotypes were identified, we defined the most common haplotype found on chromosomes I, IV, V, and X as the swept haplotype. We then retained the swept haplotypes within isotypes that passed the following per chromosome filters: total length >1 Mb; total length/maximum population-wide swept haplotype length >0.03. We classified chromosomes within isotypes as swept if the sum of the retained swept haplotypes for a chromosome was >3% of the maximum population wide swept haplotype length for that chromosome.

## Environmental parameter analysis

We calculated the pairwise distances among all *C. elegans*-positive collections on Hawaii and detected five distinct geographic clusters, each of which contain collections that are within 20 meters of one another. The largest of these clusters comprised 18 collections in the Kalopa State Recreation Area on the Big Island of Hawaii. This cluster contained 11 collections from gridsect-3 and seven additional collections within 20 meters from the edge of the gridsect. The other four geographic clusters contain four or fewer collections each. We used the average values of environmental parameters from geographically clustered collections to avoid biasing our results by local oversampling. We applied this strategy to the comparison of environmental parameters between the Hawaiian admixture populations and used the Kruskal-Wallis test to detect differences ($\alpha = 0.05$).

## Climate data analysis

Climate data were downloaded from the Integrated Surface Data (ISD) FTP server (ftp://ftp.ncdc.noaa.gov/pub/data/noaa/) managed by the National Oceanic and Atmospheric Administration (NOAA), and the National Climatic Data Center (NCDC). The data were processed using the noaa-nf pipeline available at the GitHub repository https://github.com/AndersenLab/noaa-nf (*Evans and Crombie, 2019*; copy archived at https://github.com/elifesciences-publications/noaa-nf).

# Acknowledgements

We thank the members of the Andersen lab for editing the manuscript for flow and content and for making reagents used in the experiments presented. We are grateful to landowners who gave us permission to collect nematodes on their property. We also thank individuals who have helped us collect additional strains. We would also like to thank the Hawaii Department of Land and Natural Resources as well as the Natural Area Reserves System for permitting, support for these studies, and general advice about the Hawaiian Islands. Additionally, Dr. Sam Gon from The Nature Conservancy Hawai'i Program helped with the naming of *Caenorhabditis oiwi*. This research was supported by start-up funds from Weinberg College of Arts and Sciences and the Molecular Biosciences department. KK is supported by NSF DEB 0922012 to DHA Fitch.

# Additional information

## Funding

| Funder | Grant reference number | Author |
| --- | --- | --- |
| National Science Foundation | DEB 0922012 | Karin Kiontke |

The funders had no role in study design, data collection and interpretation, or the decision to submit the work for publication.

## Author contributions

Tim A Crombie, Data curation, Formal analysis, Visualization, Methodology; Stefan Zdraljevic, Daniel E Cook, Conceptualization, Data curation, Formal analysis, Visualization, Methodology; Robyn E Tanny, Conceptualization, Formal analysis, Methodology; Shannon C Brady, Kathryn S Evans, Steffen Hahnel, Daehan Lee, Briana C Rodriguez, Conceptualization, Formal analysis, Visualization, Methodology; Ye Wang, Data curation, Formal analysis; Gaotian Zhang, Joost van der Zwagg, Methodology; Karin Kiontke, Formal analysis, Visualization, Methodology; Erik C Andersen, Conceptualization, Formal analysis, Supervision, Funding acquisition, Methodology, Project administration

## Author ORCIDs

Tim A Crombie ⬤ https://orcid.org/0000-0002-5645-4154
Stefan Zdraljevic ⬤ http://orcid.org/0000-0003-2883-4616
Karin Kiontke ⬤ http://orcid.org/0000-0003-1588-4884
Erik C Andersen ⬤ https://orcid.org/0000-0003-0229-9651

**Decision letter and Author response**
Decision letter https://doi.org/10.7554/eLife.50465.sa1
Author response https://doi.org/10.7554/eLife.50465.sa2

## Additional files

### Supplementary files
• Source data 1. Complete sampling data for the collections and isolations completed for this study as a .csv file.

• Source data 2. Cryopreserved strains that are available upon request, including any known alternative names that might be used in the literature.

• Supplementary file 1. Collection categories identified on each island. Multiple collection categories were found for some samples. For this reason, the total number of distinct collections (2,594) exceeds the total number of samples (2,263).

• Supplementary file 2. Description of variant sets used in this study. The additional filters applied to the PopGen VCF for specific uses are described in methods.

• Supplementary file 3. Nematode field sampling data form.

• Supplementary file 4. Nematode isolation data form.

• Supplementary file 5. Nextflow pip elines used in our study.

• Transparent reporting form

### Data availability
All data generated or analyzed during this study are included in the manuscript and supporting files. Source data are provided for all Figures.

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

## Appendix 1

### *Caenorhabditis oiwi* new species description

The electronic edition of this article conforms to the requirements of the amended International Code of Zoological Nomenclature (ICZN), and hence the new names contained herein are available under that Code from the electronic edition of this article. This published work and the nomenclatural acts it contains have been registered in ZooBank, the online registration system for the ICZN. The ZooBank LSIDs (Life Science Identifiers) can be resolved and the associated information viewed through any standard web browser by appending the LSID to the prefix 'http://zoobank.org/'. The LSID for this publication is: urn:lsid:zoobank.org:pub:DBB22717-DA3E-4B44-A381-DDD89B4CF1BD. The electronic edition of this work was published in a journal with an ISSN.

### *Caenorhabditis oiwi* Crombie et al. *sp. n.*

We isolated and identified a new *Caenorhabditis* species that we named *Caenorhabditis oiwi* sp. n. for the Hawaiian word meaning native. Here, we justify the species status of *C. oiwi* sp. n. based on molecular barcodes and biological species inference from mating experiments. The type isolate for *C. oiwi* sp. n. is strain ECA821. We also made an isogenized version of ECA821 by ten generations of sib mating (named ECA1100). The species reproduces sexually with males and females. The ITS2 sequence from ECA1100 *C. oiwi* sp. n. (Genbank Accession: MN056420) differs from that of all previously described *Caenorhabditis* species for which such information is available (*Félix et al., 2014*; *Ferrari et al., 2017*; *Huang et al., 2014*; *Kiontke et al., 2011*; *Slos et al., 2017*; *Stevens et al., 2019*). Note that these ribosomal DNA sequences might vary slightly within the species. Based on molecular data, *C. oiwi* sp. n. falls into the *Elegans* supergroup of *Caenorhabditis* (*Kiontke et al., 2011*) with the closest known species being *C. kamaaina* (*Félix et al., 2014*). Reciprocal mating experiments of *C. oiwi* sp. n. ECA821 with the *C. kamaaina* type isolate QG122 did not yield any viable progeny. *C. kamaaina* was previously described as a sister species to the *Japonica* group but was recently placed as the most basally diverging species in the *Elegans* group (*Kiontke et al., 2011*; *Stevens et al., 2019*). The discovery of *C. oiwi* sp. n. might help with resolving the shifting topology in this part of the *Caenorhabditis* phylogenetic tree.

The type isolate ECA821 was collected in August of 2017 from the Island of Oahu, Hawaii (21.33611°N, -157.7999°W) where it was isolated from a cluster of freshly fallen flowers. ECA821 is deposited as a cryo-preserved living stock at the *Caenorhabditis* Genetics Center. Isolate ECA821 is deposited in the NYU Rhabditid Collection and was used to study the morphology of the species (*Figure 1—figure supplement 1*; *Figure 1—figure supplement 2*). In agreement with the similarity of their rRNA sequences, *C. oiwi* sp. n. and *C. kamaaina* are at present morphologically indistinguishable. Both species show the common features of the *Elegans* group of *Caenorhabditis* (*Sudhaus and Kiontke, 2007*). Their lips are separate; the stoma is long and bears three flaps of moderate size at the metastegostom (*Figure 1—figure supplement 1A-B*). The male tail shows the typical heart-shaped, anteriorly closed fan (bursa) with a serrated edge and a shallow terminal notch (*Figure 1—figure supplement 2A, E*). The nine pairs of rays are arranged as is typical for the *Elegans* group with two pairs of rays positioned precloacally and the tips of ray pairs v1 are attached to the dorsal side of the fan. The anterior dorsal ray (ad) is in position five and the posterior dorsal ray (pd) in position seven. The tips of the sixth pair of rays (v5) are embedded in the cuticle. Rays v4 are much thinner and always shorter than ad, a character that distinguishes *C. oiwi* sp. n. and *C. kamaaina* from most species of the *Elegans* group (but not all; *C. doughertyi*, *C. tropicalis* and *C. nigoni* also have a narrower and shorter ray v4). Several species of the *Japonica* group show modified rays v4. In *C. japonica*, *C. nouraguensis*, *C. panamensis* and *C. waitukubuli*, rays v4 are much shorter than the ad rays. In *C. becei* and *C. macrosperma*, rays v4 are only slightly shorter than the ad rays, but not as skinny as in *C. kamaaina* and *C. oiwi*. The spicules are slender and their tip is pointed. The gubernaculum shows the usual forked distal tip and lateral

ears (*Figure 1—figure supplement 2C, D*), but both are more prominent than in most other species of the *Elegans* group. Here, only *C. inopinata* and *C. brenneri* have equally solid lateral ears and distal forked tip. The morphology of the females (*Figure 1—figure supplement 1B, F*) is in agreement with that of the stem species pattern of the *Elegans* group (*Sudhaus and Kiontke, 2007*).

