## [Decision Letter]

**Acceptance summary:**

The authors report a large collection of *Caenorhabditis* nematodes from Hawaii combined with worldwide samples. They document ecological and environmental differences between the different species, which provides an important ecological context for the model species *C. elegans*. Sequencing of *C. elegans* reveals that their genetic diversity in Hawaii exceeds levels of diversity seen previously in the world-wide populations, suggesting that Hawaii and the Pacific may represent an ancestral range (or at least a repository of ancestral diversity). These Hawaiian samples also lack, for the most part, the large-scale selective sweeps that dominate the genomes of other *C. elegans*.

**Decision letter after peer review:**

Thank you for submitting your article "Deep sampling of Hawaiian *Caenorhabditis elegans* reveals high genetic diversity and admixture with global populations" for consideration by *eLife*. Your article has been reviewed by two peer reviewers, and the evaluation has been overseen by a Reviewing Editor and Diethard Tautz as the Senior Editor. The following individual involved in review of your submission has agreed to reveal their identity: Asher Cutter (Reviewer #2).

The reviewers have discussed the reviews with one another and the Reviewing Editor has drafted this decision to help you prepare a revised submission.

Essential revisions:

Drawing from reviewer 1 we all agreed that the paper should acknowledge the current limitations of sampling more, i.e. it may be that the Pacific region at large shows a high diversity and there is nothing special about Hawaii.

The reviewers' full reviews are given below. The points they raise are reasonable, and I think can be easily be addressed mostly just with some rewriting.

*Reviewer #1:*

This manuscript reports on a large collection effort of *Caenorhabditis* nematodes in the Hawaiian islands (>2000 samples). The authors isolate five *Caenorhabditis* species from about 150 samples: *C. briggsae, C. elegans, C. tropicalis, C. oiwi* (a new species described here) and *C. kamaaina. C. elegans* is found at cooler temperatures and higher altitude than the others. The authors sequence the genomes of all *C. elegans* (their isolates and prior ones from Hawaii) and analyze the single-nucleotide variant diversity. The molecular diversity of *C. elegans* is three-fold higher in the Hawaii set than in the present set coming from the rest of the world. The within-Hawaiian population structure is altitudinal. Admixture analysis suggests that there is some gene flow between low-altitude Hawaiian populations and the rest of the world. The authors discuss Hawaii as a center of diversity and potential origin of the species.

The results are important and considerably extend the known diversity of this model organism, as well as shedding light on its evolutionary history. I see no major flaw.

A few suggestions to improve the manuscript:

1) It may be that the Pacific region at large shows a high diversity and there is nothing special about Hawaii.

The present sampling is on the Hawaiian islands. Prior sampling is mostly in North America and Europe, and little in the Pacific region, which introduces a bias. This should be made clearer, for example in the second paragraph on subsection “Hawaiian *C. elegans* are divergent from the global population” and in the Discussion. The Discussion is centered on the center of origin of *C. elegans* and the supposed ancestral niche. Although the Pacific region is mentioned, it would be good to more systematically indicate that one cannot distinguish at the moment whether this diversity is particular to the Hawaiian islands. That *C. inopinata* was found in South Japan could be mentioned.

Second paragraph of subsection “Hawaiian *C. elegans* are divergent from the global population”: note that CeNDR shows more strains with the corresponding "Hawaiian-like" isotypes (BRC20231 from Taiwan (?), JU1171 from Chile, thus from the Pacific region). In line with this, it could be noted in the penultimate paragraph of the Results section that the admixed isotype was QG556 (and QG558), from California, also on the Pacific rim.

2) Could it be that the relatively high diversity is due to a lack of selective sweeps reducing the diversity (due to the high linkage disequilibrium) in the Hawaii set because of geographic isolation? Diversity may have been higher elsewhere before and lost perhaps due to human activity and movement.

3) Gene flow between "Hawaii/Pacific" and the rest of the world.

I do not understand why there is emphasis on the gene flow with the C group and little is made (i) of the gene flow to the G group that is shown in Figure 5—figure supplement 2 and Figure 7—figure supplement 1; and (ii) of the admixture with the “HI Low” of some non-Hawaiian strains (including the G group). It seems that both “HI Low” and “HI Invaded” are partially admixed.

4) The category of “fruits, nuts or vegetation”, opposed to “flower” is odd. First, nuts are fruits. Second, flower is vegetation. Please at least change the name and perhaps reconsider the categories.

5) I could not find the correspondence between the samples, the strains, the species, the isotypes. The supplementary table with the sampling detail lacks the strain and species names in each sample.

6) The Discussion is currently centered on the center of origin of *C. elegans*. However, some other points could be discussed, for example the use of these data and strains as resource, e.g. for GWAS, the sampling design, the other species, etc.

*Reviewer #2:*

Crombie et al. present the collection of and population genomic analysis for the largest such effort for *C. elegans*, and *Caenorhabditis* more generally. They focus their study on ~100 samples isolated and whole-genome sequenced from the Hawaiian islands, and integrate this outstanding dataset with ~175 *C. elegans* wild isolate strains from around the world that the authors genome-sequenced previously. Their data collection also incorporates an extremely streamlined protocol for associating geographic, ecological and environmental data with the isolates, which they use to demonstrate species differentiation by elevation/temperature (in the process discovering and naming a new species) as well as some genetic differentiation within *C. elegans* by island and elevation. They corroborate some previously established patterns of polymorphism across the genome and, more novel, discover a trove of diversity for those isolates from Hawaii. The data and analysis pipeline are publicly available. This abundance of polymorphism in Hawaii exceeds the sum total of molecular variation for all other strains found around the world and hints at the evolutionary and ecological origins of *C. elegans* as a species potentially deriving from cool, high-elevation habitats in or around the Pacific within the last 10My.

The clearly presented analysis, despite comprising an impressively large sample of genomes, is relatively simple. These results will be valuable for the *C. elegans* research community, especially for researchers interested in its ecology and evolutionary history, but also those interested in using natural allelic variants to understand molecular mechanisms. My criticisms are mostly to do with wording, specifically around the meaning and consistency of use of "population."

Subsection “Hawaiian *C. elegans* are divergent from the global population” and elsewhere: use of the word "population" to refer to the Hawaiian vs. non-Hawaiian samples seems a bit inconsistent/misleading, as the additional analyses show that they each comprise multiple genetic clusters (also referred to as populations) and the non-Hawaiian sample in particular is geographically global, and so may confuse readers. I would be more comfortable with "population sample" or "sample of strains" or just "sample." This issue appears again in paragraph two of the Discussion after much effort went into analyzing distinct genetic groups within the Hawaii sample that are not entirely defined by geography. In subsection “The ancestral niche of *C. elegans* might be similar to the Hawaiian niche”, it seems the use of "populations" might better be replaced by "localities." Please comb through the manuscript for use of "population" to ensure consistency in what is meant, and what the analysis shows.

Subsection “*C. elegans* population structure on Hawaii”: I think this conclusion may be too strong. The admixture analysis does provide evidence for 11+ genetically-distinguishable groups, though whether these necessarily correspond to (geographic) ancestral populations is tricky to infer. This challenge is especially true given the highly selfing reproduction of *C. elegans* that violates assumptions for what populations mean in admixture, so some caution is needed in interpretation. I'd be more comfortable with wording along the lines of, "This analysis identified at least 11 genetic groups for this sample of *C. elegans*, each of which we refer to as a population for the purposes of further analysis."

---

## [Author Response]

Essential revisions:Drawing from reviewer 1 we all agreed that the paper should acknowledge the current limitations of sampling more, i.e. it may be that the Pacific region at large shows a high diversity and there is nothing special about Hawaii.

Yes, we completely agree about this concern. In the Results (“Hawaiian *C. elegans* are divergent from most strains sampled across the globe” and “*C. elegans* population structure”) and the Discussion (“The origins of *C. elegans*”) we have added content to address the sampling biases in Hawaii and Europe and more systematically indicate that we cannot know whether this diversity is specific to the Hawaiian Islands.

The reviewers' full reviews are given below. The points they raise are reasonable, and I think can be easily be addressed mostly just with some rewriting.Reviewer #1:[…]A few suggestions to improve the manuscript:1) It may be that the Pacific region at large shows a high diversity and there is nothing special about Hawaii.The present sampling is on the Hawaiian islands. Prior sampling is mostly in North America and Europe, and little in the Pacific region, which introduces a bias. This should be made clearer, for example in the second paragraph on subsection “Hawaiian C. elegans are divergent from the global population” and in the Discussion. The Discussion is centered on the center of origin of C. elegans and the supposed ancestral niche. Although the Pacific region is mentioned, it would be good to more systematically indicate that one cannot distinguish at the moment whether this diversity is particular to the Hawaiian islands. That C. inopinata was found in South Japan could be mentioned.

Thank you for pointing out this limitation. In the Results and the Discussion we have added content to address the sampling biases in Hawaii and Europe and more systematically indicate that we cannot know whether this diversity is specific to the Hawaiian Islands.

Second paragraph of subsection “Hawaiian C. elegans are divergent from the global population”: note that CeNDR shows more strains with the corresponding "Hawaiian-like" isotypes (BRC20231 from Taiwan (?), JU1171 from Chile, thus from the Pacific region). In line with this, it could be noted in the penultimate paragraph of the Results section that the admixed isotype was QG556 (and QG558), from California, also on the Pacific rim.

These are excellent points. We have reworked the manuscript to highlight the Pacific region isolations for the MY23 isotype. In paragraph three of subsection “*C. elegans* population structure” we added this detail about QG556 and a new figure (Figure 6—figure supplement 1) that more clearly shows admixture and geographical proximity between the Hawaii Invaded and non-Hawaiian C populations.

2) Could it be that the relatively high diversity is due to a lack of selective sweeps reducing the diversity (due to the high linkage disequilibrium) in the Hawaii set because of geographic isolation? Diversity may have been higher elsewhere before and lost perhaps due to human activity and movement.

We added this possible explanation for the high genetic diversity found among the Hawaiian isotypes (subsection “The origins of *C. elegans*”).

3) Gene flow between "Hawaii/Pacific" and the rest of the world.I do not understand why there is emphasis on the gene flow with the C group and little is made (i) of the gene flow to the G group that is shown in Figure 5—figure supplement 2 and Figure 7—figure supplement 1; and (ii) of the admixture with the “HI Low” of some non-Hawaiian strains (including the G group). It seems that both “HI Low” and “HI Invaded” are partially admixed.

Thank you for this valuable feedback. We have added text to the Results (subsection “*C. elegans* population structure”) to reflect the evidence of gene flow from a population related to the Hawaiian Invaded and Hawaiian Low populations to the non-Hawaiian G population. We have also added to the Results (final paragraph) regarding the haplotype sharing between the Hawaiian Low and non-Hawaiian populations. Furthermore, we specifically mention that the Hawaii Low population contains some admixture with non-Hawaiian isotypes and shows some evidence of globally swept haplotypes (subsection “The ancestral niche of *C. elegans* might be similar to the Hawaiian niche”).

4) The category of “fruits, nuts or vegetation”, opposed to “flower” is odd. First, nuts are fruits. Second, flower is vegetation. Please at least change the name and perhaps reconsider the categories.

We agree that the substrate category names in the original manuscript were potentially confusing. We have changed the names and reorganized the samples into new categories. After changing the names and categories, we reran all statistical calculations involving substrate categories.

1) We removed the eight “vegetation” samples from the original “fruits, nuts or vegetation” category and recategorized them into a new “vegetation” category.

2) We recategorized all the samples originally called “other” into the new vegetation category because these samples all consisted of various forms of vegetation that were each infrequently sampled.

3) We changed the name of the original “Fruit/nut/veg” category to “Fruit” because nuts are fruits, as pointed out.

We now include our logic for choosing the six major substrate classes in the Materials and methods section: “On these hikes, we opportunistically sampled substrates known to harbor *C. elegans*,including fruits, flowers, stems, leaf litter, fungus, compost, wood, live arthropods, and molluscs (Ferrari et al., 2017; Frézal and Félix, 2015; Schulenburg and Félix, 2017). We chose to group these substrates into six major categories: leaf litter, fruit, flower, fungus, invertebrate, and vegetation. The vegetation category contains all substrates that do not fall into any of the other five categories. We chose to distinguish the leaf litter, fruit, and flower substrates from the vegetation category because we suspect these substrates provide distinct microhabitats and each substrate was sampled extensively.”

Importantly, while we were updating the categories, we discovered 19 samples that had been inadvertently filtered from our data set prior to plotting Figure 2 and running the statistics involving substrate categories in the original manuscript. These samples have been added back into the data set for plotting and analysis. No inferences from our statistical tests changed because of the reorganization or the reintroduction of the 19 missing samples.

5) I could not find the correspondence between the samples, the strains, the species, the isotypes. The supplementary table with the sampling detail lacks the strain and species names in each sample.

The sample, strain, species, and isotype names are all included in Source data 1. The column names for each are; c_label, strain, species_id, and isotype. We have updated the text to reference Source data 1 (full collection data) specifically, rather than just Supplementary file 1 (table of collection categories by island), which might have been confusing.

6) The Discussion is currently centered on the center of origin of C. elegans. However, some other points could be discussed, for example the use of these data and strains as resource, e.g. for GWAS, the sampling design, the other species, etc.

We agree that the utility of these data in GWA should be addressed in the discussion. We added text within the first paragraph of the Discussion to highlight both the utility and potential disadvantages of these data with respect to GWA (First paragraph).

Reviewer #2:[…] My criticisms are mostly to do with wording, specifically around the meaning and consistency of use of "population."Subsection “Hawaiian C. elegans are divergent from the global population” and elsewhere: use of the word "population" to refer to the Hawaiian vs. non-Hawaiian samples seems a bit inconsistent/misleading, as the additional analyses show that they each comprise multiple genetic clusters (also referred to as populations) and the non-Hawaiian sample in particular is geographically global, and so may confuse readers. I would be more comfortable with "population sample" or "sample of strains" or just "sample." This issue appears again in paragraph two of the Discussion after much effort went into analyzing distinct genetic groups within the Hawaii sample that are not entirely defined by geography. In subsection “The ancestral niche of C. elegans might be similar to the Hawaiian niche”, it seems the use of "populations" might better be replaced by "localities." Please comb through the manuscript for use of "population" to ensure consistency in what is meant, and what the analysis shows.

Thank you for this helpful criticism. We revised use of the word “population” throughout the manuscript to avoid confusion and ensure consistency with the results of our analyses. We no longer use the word “population” to refer to the Hawaiian and non-Hawaiian isotype samples; we now refer to them as “samples”. The term “Hawaiian population” has been removed. We also removed the use of the word “populations”. We changed this to “Hawaiian sampling locations”.

Subsection “C. elegans population structure on Hawaii”: I think this conclusion may be too strong. The admixture analysis does provide evidence for 11+ genetically-distinguishable groups, though whether these necessarily correspond to (geographic) ancestral populations is tricky to infer. This challenge is especially true given the highly selfing reproduction of C. elegans that violates assumptions for what populations mean in admixture, so some caution is needed in interpretation. I'd be more comfortable with wording along the lines of, "This analysis identified at least 11 genetic groups for this sample of C. elegans, each of which we refer to as a population for the purposes of further analysis."

We thank the reviewer for this helpful comment. We have updated the admixture Results section (subsection “*C. elegans* population structure”) to include the suggested text. Throughout the manuscript, we are consistent with our usage of the term “population” to refer to genetically distinct groups identified by ADMIXTURE.